# Globally distributed *Myxococcota* with photosynthesis gene clusters illuminate the origin and evolution of a potentially chimeric lifestyle

Liuyang Li [1], Danyue Huang[2], Yaoxun Hu[1], Nicola M. Rudling [3], Daniel P. Canniffe [3], Fengping Wang [1,2] ✉ & Yinzhao Wang [1] ✉

Photosynthesis is a fundamental biogeochemical process, thought to be restricted to a few bacterial and eukaryotic phyla. However, understanding the origin and evolution of phototrophic organisms can be impeded and biased by the difficulties of cultivation. Here, we analyzed metagenomic datasets and found potential photosynthetic abilities encoded in the genomes of uncultivated bacteria within the phylum *Myxococcota*. A putative photosynthesis gene cluster encoding a type-II reaction center appears in at least six *Myxococcota* families from three classes, suggesting vertical inheritance of these genes from an early common ancestor, with multiple independent losses in other lineages. Analysis of metatranscriptomic datasets indicate that the putative myxococcotal photosynthesis genes are actively expressed in various natural environments. Furthermore, heterologous expression of myxococcotal pigment biosynthesis genes in a purple bacterium supports that the genes can drive photosynthetic processes. Given that predatory abilities are thought to be widespread across *Myxococcota*, our results suggest the intriguing possibility of a chimeric lifestyle (combining predatory and photosynthetic abilities) in members of this phylum.

Photosynthesis is an ancient biological process in which light energy is transformed into metabolic energy and reducing power for carbon dioxide fixation to produce biomass[1]. The light utilization of phototrophs expands their ecological niches, facilitates their cosmopolitan distribution, contributes significantly to global biogeochemical cycles, and completely altered the Earth's surface and atmospheric environment during the Archaean and Proterozoic eons[2]. Bacterial chlorophototrophs include the oxygenic *Cyanobacteria* harboring both type I and type II photosynthetic reaction centers (RCI/II) like those found in eukaryotic plants and algae, as well as anoxygenic phototrophic prokaryotes that utilize a single RCI or RCII for light energy conversion[3].

It has been suggested that all RCs evolved from an anoxygenic common ancestor, living in an environment devoid of oxygen at -3.5–3.2 Ga[3]. Multiple lines of evidence indicate that ancestral phototrophs evolved via a complex path to form diverse phototrophs and metabolisms[4]. Due to their ecological and evolutionary importance, a great deal of effort has been made to unravel the diversity of phototrophic microorganisms[5]. To date, prokaryotes using (bacterio)chlorophyll-based phototrophy have been found in eight bacterial phyla, namely *Cyanobacteria* (-200 years ago)[6], *Proteobacteria* (-180 years ago)[7], *Bacteroidota* (the class *Chlorobia*, 1906)[8], *Chloroflexota* (1974)[9], *Firmicutes* (1983)[10], *Acidobacteriota* (2007)[11], *Gemmatimonadota* (2014)[12], and

[1]State Key Laboratory of Microbial Metabolism, School of Life Sciences and Biotechnology, Shanghai Jiao Tong University, Shanghai 200240, China. [2]School of Oceanography, Shanghai Jiao Tong University, Shanghai 200030, China. [3]Institute of Systems, Molecular and Integrative Biology, University of Liverpool, Liverpool L69 7ZB, UK. ✉e-mail: fengpingw@sjtu.edu.cn; wyz@sjtu.edu.cn

the recently discovered (2018) and cultured (2022) *Eremiobacterota*[13,14]. Each lineage contains a specific solar energy conversion device that varies in types of RC and antenna, as well as pigments biosynthetic pathways[15], by which the energy is finally used for either autotrophic or heterotrophic growth, e.g., in photoautotrophic *Cyanobacteria* or in photoheterotrophic *Firmicutes*.

*Myxococcota* is an astonishing bacterial phylum because of their extraordinary social lifestyle (e.g., predation and fruiting body formation), which is unusual in the prokaryotes[16,17]. They were originally known as *Myxobacteria* affiliated within the class *Deltaproteobacteria*[18], but were recently reclassified into a separate phylum according to the Genome Taxonomy Database (GTDB)[19]. Members of *Myxococcota* have distinct metabolic and structural characteristics. Most species exhibit the abilities of adventurous motility[20,21] and social motility[22,23]. Predatory lifestyles have also been reported in many species of this phylum, and the "wolf pack" analogy has been widely used to describe their predatory behavior[24]. For instance, the model organism *Myxococcus xanthus* is able to feed on a broad range of microbes including bacteria[25] and fungi[26]. Some members are saprophytic, and can produce antimicrobials to fend off competitors, e.g., the cellulose-degrading *Sorangium cellulosum*[27]. Therefore, the capacity to produce a plethora of secondary metabolites for predation[23] or competition has made *Myxococcota* a rich source for bioactive secondary metabolites for decades besides *Actinobacteria*, *Bacillus*, and fungi[28]. Cultured *Myxococcota* have typically been isolated from (micro)-oxic soil habitats[29]. Recently, a novel group of bacterial predators[30,31], *Bradymonabacteria* (representative of the order *Bradymonadales* of *Myxococcota*), was isolated in saline environments. Additionally, metagenomic studies have further expanded our knowledge of the wide distribution of *Myxococcota* in various habitats, including freshwater[32], seawater[33], and potentially anoxic habitats such as the human gut[34]. Therefore, *Myxococcota* are cosmopolitan and considered as keystone taxa that play significant roles in microbial interaction networks by their versatile capabilities.

In this study, we performed a global census for phototrophic prokaryotes, and found that the bacterial phylum *Myxococcota* contains potentially chlorophototrophic members that have a photosynthesis gene cluster (PGC) encoding all enzymes for bacteriochlorophyll biosynthesis, reaction center and light-harvesting proteins, and other key enzymes. Results of the heterologous expression of their pigment biosynthesis genes strongly suggest that these genes can drive photosynthetic processes in their original myxobacterial background. Phylogenies of photosynthetic genes showed a monophyletic origin and vertical inheritance in the potentially phototrophic *Myxococcota*. Additionally, several members appear to have carbon fixation capacity with a complete Calvin-Benson-Bassham (CBB) cycle. Thus, our results indicate that the phylum *Myxococcota* has an ancient phototrophic history, and suggest that phototrophic *Myxococcota* may be playing overlooked roles in ecosystems.

## Result and discussion
### Discovery of potentially phototrophic *Myxococcota*
In order to provide a universal overview of potential phototrophs in prokaryotes, we first constructed a manually curated reference protein database with RCs, photosynthetic pigment biosynthesis enzymes, regulatory proteins, and antenna systems. We subsequently assembled genomes from public metagenomes and downloaded genomes from the National Center for Biotechnology Information (NCBI) prokaryotes database and the Earth's Microbiome (GEM) catalog. Then genomes were used as queries for screening photosynthesis related homologs (identity > 30%, $e < 1 \times 10^{-20}$, coverage >75%) and the results were manually checked to identify potential phototrophs. A total of 8221 genomes containing a partial to complete (gene prevalence >50%) (bacterio)chlorophyll biosynthesis pathway were retained. Among

these genomes, we found 32 MAGs that phylogenetically belong to the phylum *Myxococcota* and are thus placed outside the eight known phototrophic phyla, according to the taxonomy assignment by GTDB-Tk[35] (Supplementary Data 1). All 32 MAGs contain genes coding for RCII (e.g., *pufLM*), photosynthetic pigment biosynthesis, regulatory proteins, and antenna systems, strongly suggesting chlorophototrophy in *Myxococcota* (Supplementary Data 2). The completeness of these 32 MAGs is >70% according to Gradient Boost Model in CheckM2[36], whereas the contamination of nearly all MAGs is below 5% (Supplementary Data 1).

We clustered these 32 MAGs on the basis of 95% whole-genome average nucleotide identities (ANIs) to uncover species-level diversity, revealing 18 candidate species-level taxa (Fig. S1 and Supplementary Data 1). To confirm the taxonomic relationship of these MAGs containing (near) complete sets of phototrophic genes, a species tree consisting of the 32 MAGs and reference bacterial genomes from the NCBI genome database was constructed based on a concatenated set of 37 marker genes (Fig. 1a and Supplementary Data 3). Reference genomes from the GTDB r207 (Supplementary Data 4 and 5) were also used to construct species trees (Figs. S1–S3) based on the concatenated alignment of 120 housekeeping genes[35]. These trees consistently determine that the *Myxococcota* MAGs with photosynthesis genes distribute in at least six discrete lineages classified into 14 genera and are interspersed with non-phototrophic members within this phylum. Specifically, three MAGs form a deep-branching cluster that shares ~45–49 whole-genome average amino acid identities (AAIs) and ~61–72 ANIs with other myxococcotal classes (Fig. S1). We proposed the name *Candidatus* (*Ca.*) Kuafubacteria for this class-level lineage (placeholder name c_WYAZ01 in GTDB r207), from the sun-chasing giant Kua Fu in ancient Chinese mythology. Eight MAGs cluster within the family *Myxococcaceae* from the class *Myxococcia* (Fig. S2), and 21 MAGs form four clusters within the class *Polyangia* (Fig. S3). In *Polyangia*, six MAGs form a close cluster within the family *Nannocystaceae*, three MAGs belong to the family *Polyangiaceae*, and one MAG is clustered within the family *Sandaracinaceae*. We proposed the prefix *Xihe* (Xi He, sun goddess in Chinese mythology) combined with the environments where these MAGs were recovered to name these uncultured and/or unclassified lineages, e.g., *Ca.* Xihehalomonas phototrophica that was recovered from salt lagoon. Additionally, 11 MAGs form a cluster (f_SG8_38) within the *Polyangia*, and we tentatively named *Ca.* Houyibacteriaceae for this family-level lineage, after Hou Yi, a sun-related immortal in ancient Chinese mythology. A protologue for the proposed taxonomic names is provided in the Supplementary Note 1.

### Phylogenic inference of photosynthesis genes
In the *Proteobacteria*, *Gemmatimonadota*, and *Firmicutes*, photosynthesis related genes are clustered into a continuous stretch with a characteristic ensemble of operons, named the PGC. A total of 1845 PGCs were retained from the 8221 potential phototrophic genomes. These PGCs were taxonomically affiliated within the phyla *Myxococcota*, *Proteobacteria* and *Gemmatimonadota*, while the PGCs of *Firmicutes* were excluded due to their short length and drastic difference with classic PGCs. Remarkably, for *Myxococcota*, all of the six families contain PGCs coding for all proteins of RCII, enzymes of bacteriochlorophyll and carotenoid biosynthesis, and regulatory proteins (Fig. 2). The phylogenies of PGC genes can provide robust evolutionary inference[37], and here we constructed phylogenetic trees of the PGCs, as well as photosynthetic reaction center proteins (PufLM)[38], magnesium chelatase encoded by *bchHDI* genes[39], and light-independent protochlorophyllide reductase encoded by *bchLNB* genes[40]. Phylogenetic placements of the PGC, PufLM, BchHDI, and BchLNB trees all indicate that *Myxococcota* phototrophic genes are monophyletic but cluster within the proteobacterial clades (Figs. 1b and 2c, and Figs. S4 and S5).

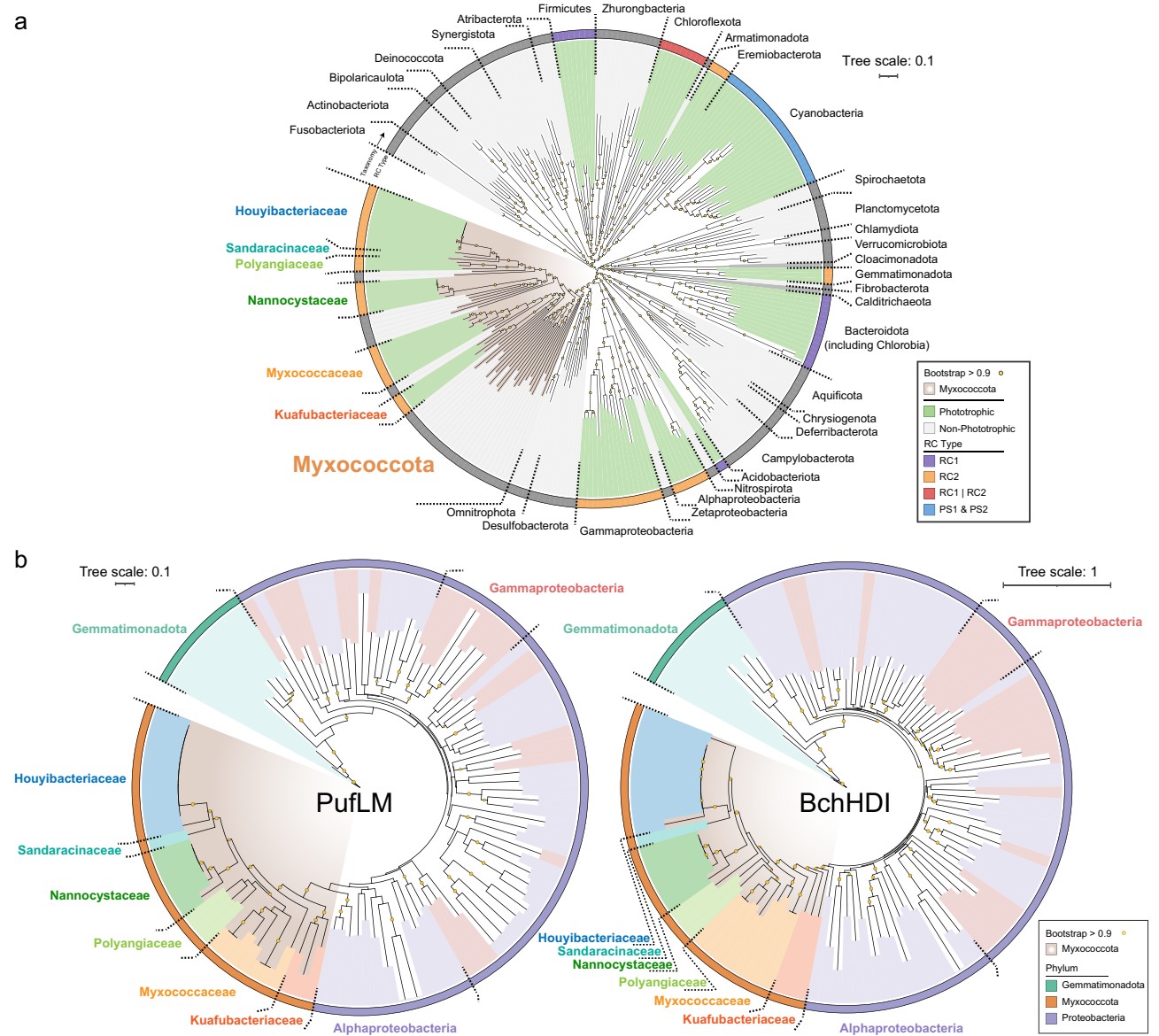

**Fig. 1 | Phylogenetic affiliations of the potential phototrophic _Myxococcota_ MAGs, and corresponding PufLM and BchHDI protein sequences.**
**a** Phylogenomic affiliation of the MAGs based on 37 conserved protein sequences and using 232 representative reference bacterial genomes. Alignments were based on MAFFT and then filtered with trimAl. The tree was built using IQ-Tree with 1,000 bootstrap replicates. The bootstrap supporting values above 0.9 were indicated with solid circles. The phylogenetic tree was rooted at the midpoint. In the inner ring, lineages of _Myxococcota_ are shaded with brown gradient background. In the middle ring, chlorophototrophic lineages are colored with green background, and non-chlorophototrophic lineages are colored with gray background. The outer ring of phylogeny is colored based on different types of reaction centers. All phyla are labeled near the phylogenies, with the exception of _Proteobacteria_ that are labeled based on different classes, e.g., _Alphaproteobacteria_ and _Gammaproteobacteria_

(_Betaproteobacteria_ shared same color with _Gammaproteobacteria_ based on the taxonomy in GTDB r207). Six potential phototrophic families of _Myxococcota_ are also labeled with different colors near their phylogenies. **b** The phylogenetic trees were constructed based on the alignments of PufLM with 571 aligned positions and BchHDI with 1702 aligned positions. Alignments were based on MAFFT and then filtered with trimAl, and the trees were built using IQ-Tree with 1000 bootstrap replicates. The bootstrap supporting values above 0.9 are indicated with solid circles. Lineages of _Myxococcota_ are colored with brown gradient for background. Members from _Alphaproteobacteria_, _Gammaproteobacteria_ (including _Betaproteobacteria_), _Gemmatimonadota_, and six families of _Myxococcota_ are assigned different background colors. The outer ring of phylogeny is colored based on corresponding phylum. Detailed information for 32 MAGs is provided in Supplementary Data 1.

The PGC gene organization patterns were identified for these genomes (Fig. 2c). Overall, the PGC arrangements of _Myxococcota_ varies with different lineages but are still highly similar with those of _Proteobacteria_ and _Gemmatimonadota_, such as the regular gene orders of _bchIDO_, _crtCDEF-bchCXYZ_, _pufBALMC_[41]. However, the operon containing _bchP_ and _bchG_, separating by an ORF coding for a protein sharing a high identity with the light-harvesting complex assembly factor LhaA, which is commonly found in _Proteobacteria_ and _Gemmatimonadota_, is found only in the family _Ca._ Kuafubacteriaceae. The

other lineages of _Myxococcota_ display a unique pattern in which the _bchPG_ operon forms a compact arrangement by combining with the commonly occurring _bchFNBHLM_ and _pufH(puhA)-puhBC-bciE-acsF-puhE_ operons[42].

The highly similar topologies of photosynthesis gene trees (Fig. 1b, and Figs. S4–S6) and the analogous PGC arrangement (Fig. 2), strongly indicate a vertical inheritance of chlorophototrophy in _Myxococcota_. Nevertheless, as all myxococcotal photosynthesis genes branch within the _Proteobacteria_ clade on phylogenetic trees, we propose that one

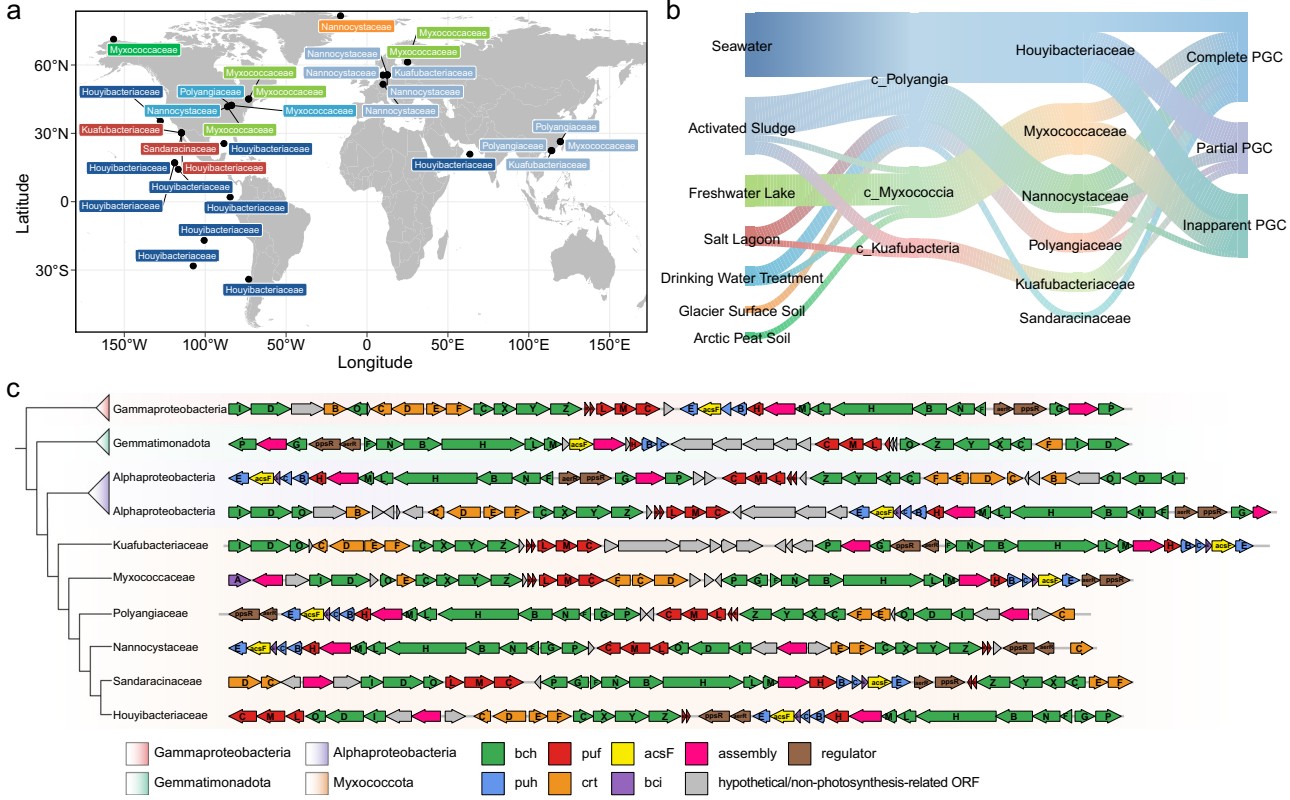

**Fig. 2 | The global distribution of potential phototrophic *Myxococcota*, and gene organizations and phylogenies of their PGCs in comparison with *Proteobacteria* and *Gemmatimonadota*. a** The cosmopolitan distribution of the photosynthesis gene-containing *Myxococcota*. Each genome is colored by its corresponding habitat and labeled with its taxonomic affiliation near corresponding geographical location. Fourteen MAGs (average completeness ~92.6%) possess (near) complete PGCs with 26-35 genes coding for all proteins of the RCs, enzymes of bacteriochlorophyll and carotenoid biosynthesis, regulatory proteins, and cofactors. Eight MAGs (average completeness ~85.3%) possess partial PGCs, consisting of either only one cluster with 12-24 photosynthesis genes or fragmented clusters dispersed in different contigs. Photosynthesis genes did not form apparent PGC (photosynthesis gene number <12 in any clusters) for the remaining 10 MAGs (average completeness ~84.6%), potentially attributed to incomplete assembly and/ or binning of fragment reads in the next generation sequencing. **b** Sankey diagram shows the relationship between habitats and taxonomy of the photosynthesis gene-

containing *Myxococcota*. **c** Gene organizations and phylogenies of representative myxococcotal PGCs. Genomic data was referenced for the PGC composition: the most closely related alphaproteobacterial genomes, *Algihabitans albus* HHTR 118 (GenBank accession no. GCA_003572205.1) and the genome belonging to the family *Inquilinaceae* (GEM accession no. 3300026561_2), the distantly related *Gemmatimonadota* genome *Gemmatimonas* sp. (GenBank accession no. GCA_020247115.1) and gammaproteobacterial genome *Methyloversatilis discipulorum_A* (GenBank accession no. GCA_018780585.1). Members from *Alphaproteobacteria*, *Gammaproteobacteria*, *Gemmatimonadota*, and *Myxococcota* are assigned different background colors. Photosynthesis genes are also colored: *bch* (green) and *bci* (purple), bacteriochlorophyll biosynthesis genes; *puf* (red), genes encoding reaction center proteins; *puh* (blue), genes encoding reaction center assembly proteins; *crt* (orange), carotenoid biosynthesis genes; gray, hypothetical genes or those of no relevance to phototrophy. Detailed information for PGC is provided in Supplementary Data 2.

ancient *Myxococcota* lineage obtained an entire PGC from an early proteobacterium, leading to the origin of phototrophy of *Myxococcota*. The distribution of phototrophic lineages across its species trees is wide, with phototrophic clades scattered across 14 genera in six families from five orders in three classes of *Myxococcota* (Fig. 1a, and Fig. S1). It has been suggested that the compact arrangement of the PGC may facilitate horizontal transfer[43], or be convenient for efficient regulation and gene expression[44]. It is unlikely that the 14 genera of the phylum *Myxococcota* independently laterally acquired the PGC multiple times from one lineage of phototrophic *Proteobacteria*. On the contrary, it may be best explained that the common ancestor of these *Myxococcota* lineages acquired phototrophic ability, with further multiple independent losses in other lineages (regressive evolution). In addition, it is widely accepted that the origin of the eukaryotic mitochondrion was an endosymbiotic process that involved the predation of an alphaproteobacterium. Therefore, it is likely that phototrophy in *Myxococcota* either (i) originated from an ancient acquisition by predation-based HGT from a proteobacterium, during which the entire PGC was integrated into the genome of a predatory ancestor, or (ii) originated from the 'classical' HGT via acquisition of a mobile genetic

element, the scenario proposed for *Gemmatimonadota*[12]. Alternatively, as a recently reclassified phylum, *Myxococcota* were previously assigned to the class *Deltaproteobacteria*[19]. Thus, we also cannot rule out the possibility that photosynthetic ability was present in the last common ancestor of *Proteobacteria* and *Myxococcota*, resembling previous suggestions that photosynthesis was a common trait of all bacteria or a broad clade including all extant phototrophs[45], followed by extensive losses in most lineages.

In all photosynthesis gene trees, the family *Ca*. Kuafubacteriaceae branched as an early diverging member within the phylum *Myxococcota* neighboring with the alphaproteobacterial lineage. The family *Myxococcaceae* contains two clades in the PufLM and BchHDI trees, with one clade neighboring with the *Ca*. Kuafubacteriaceae and the other clade located within the class *Polyangia*. The families *Nannocystaceae*, *Polyangiaceae*, *Sandaracinaceae*, and *Ca*. Houyibacteriaceae of the class *Polyangia* separate clearly, indicating an apparent vertical inheritance, with the exception that one clade of *Polyangiaceae* is located near the family *Myxococcaceae* in the BchHDI and BchLNB trees. Overall, the patchy distribution of phototrophs in diverse *Myxococcota* families is most likely caused by a combination of

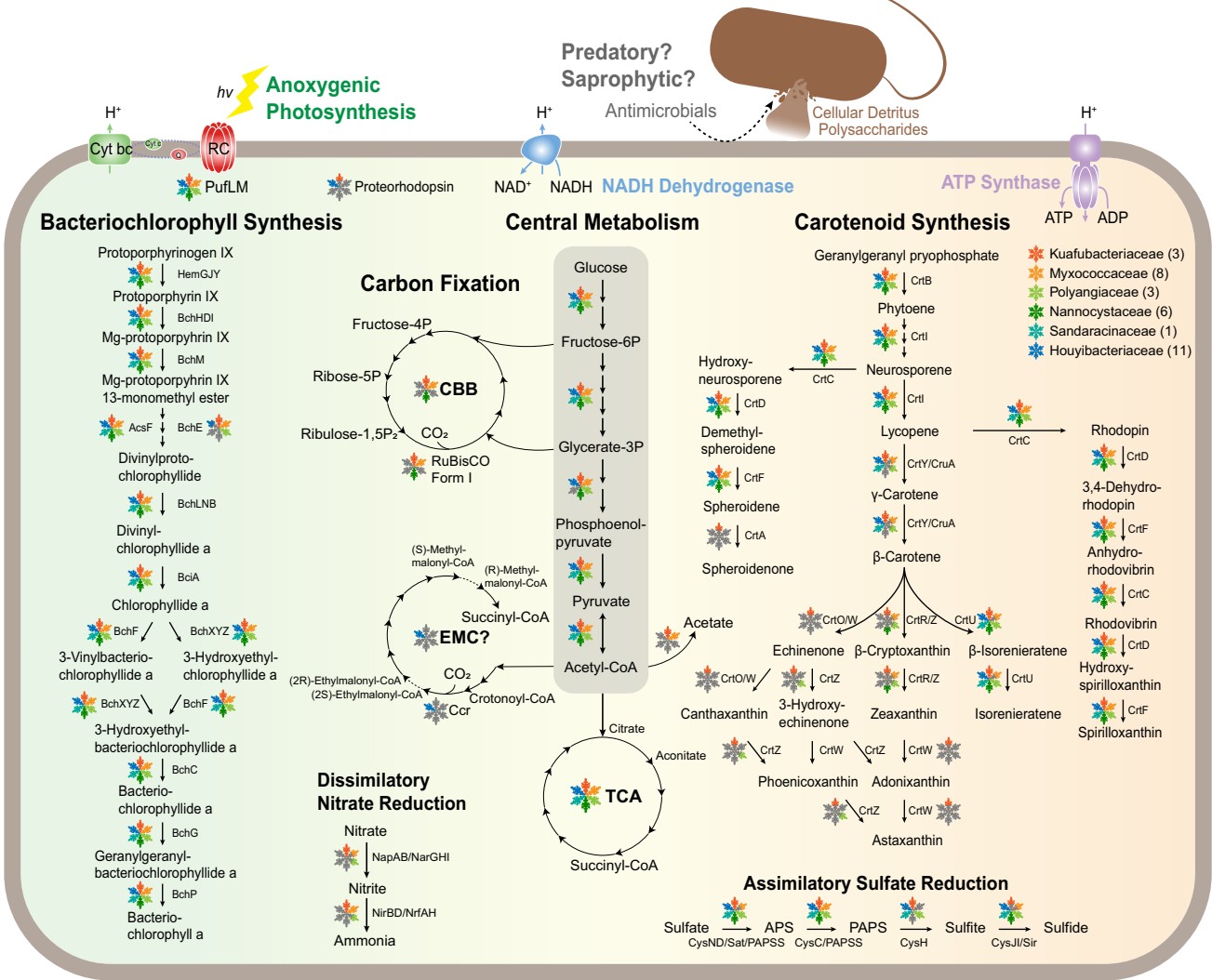

**Fig. 3 | Metabolic schemes of potential phototrophic *Myxococcota*.** Colored parts in snowflake indicate the presence/absence of metabolic pathways and proteins for representative genome of each *Myxococcota* family. Half-filled parts in snowflake indicate the absence of proteins in the representative genome but present in other genomes from the same family. Gray color denotes the pathway or protein is absent in all genomes of this family. Representative genome of the family *Nannocystaceae* contains complete CBB cycle. Representative genomes of the families *Myxococcaceae* and *Ca.* Kuafubacteriaceae contain near complete CBB cycle with key enzymes, and sole missing enzymes can be identified in other genomes from corresponding family. The following metabolic pathways were used in the figure: bacteriochlorophyll synthesis pathway; TCA cycle, citrate cycle; glycolysis pathway; CBB cycle, Calvin-Benson-Bassham cycle; EMC, ethylmalonyl-CoA pathway; assimilatory sulfate reduction; dissimilatory nitrate reduction; carotenoid synthesis pathways, e.g., spheroidene pathway, astaxanthin biosynthesis pathway, and spirilloxanthin pathway. Several marker proteins were also included in the figure, e.g., proteorhodopsin; PufLM, photosynthetic reaction center subunits; form I RuBisCO, ribulose-1,5-bisphosphate carboxylase/oxygenase. Detailed information for representative MAGs is provided in Supplementary Data 2.

vertical inheritance of photosynthetic capacities, which is also supplemented by occasional HGT events between descendant lineages.

Estimating the date of origin of phototrophy in multiple lineages may provide key clues to determine the original inventors of phototrophy. For instance, based on our estimation by molecular dating via species tree, the last common ancestor of six phototrophic families of *Myxococcota* originated at ~3.09–2.67 Ga. Meanwhile, the last common ancestor of *Proteobacteria* and *Myxococcota* originated at ~3.56–3.17 Ga, close to the estimated origin of anoxygenic photosynthesis[3]. Thus, it is reasonable to assume that ancestral *Myxococcota* serve as an old phototrophic lineage that arose before the great oxygenation event (GOE) at ~2.33 Ga[46,47]. The early origin of phototrophy in *Myxococcota*, whose crown group is currently dominated by aerobic and/or facultative anoxygenic species, reflects the adaption of anaerobic ancestors to subsequent oxic environments. However, previous efforts to explain the early evolution of photosynthesis have been restricted to

the limited phototrophic lineages known at the time[48,49]. The expansion in the number of phototrophic bacterial phyla will provide a crucial foundation for more accurate future reconstructions of the evolutionary history and relationships among phototrophs.

### Light-driven metabolisms

We found a complete pathway for the biosynthesis of bacteriochlorophyll *a* in all of the six families belonging to the phylum *Myxococcota* (Fig. 3). To test the activity of a bacteriochlorophyll biosynthesis enzyme encoded in a myxococcotal PGC, the catalytic components of the enzyme chlorophyllide oxidoreductase (COR), encoded by genes *bchYZ* were chosen for heterologous expression. The overlapping *bchYZ* from a member of *Ca.* Kuafubacteriaceae were synthesized and cloned into a purple bacterial expression vector. A mutant of the model purple bacterium *Rhodobacter* (*Rba.*) *sphaeroides* lacking the native *bchY* and *bchZ* genes, along with the 8-vinyl

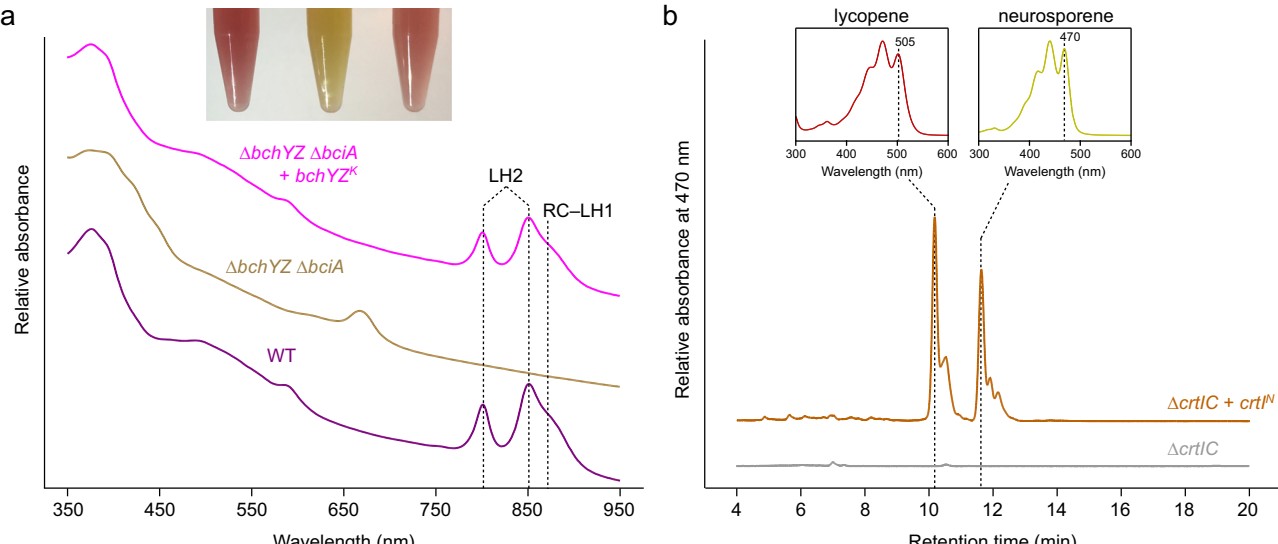

**Fig. 4 | Heterologous expression of myxococcotal genes for pigment biosynthesis. a** Whole-cell absorption spectra of strains of *Rba. sphaeroides*. The wild-type (WT) spectrum has absorption features for LH2 and RC–LH1 that are missing in the Δ*bciA* Δ*bchYZ* mutant. Heterologous expression of *bchYZ* from LLY-WYZ−16_1 (*Ca*. Kuafubacteriaceae) in this mutant background restored the pigmentation of the WT (inset), and the assembly of LH2 and RC–LH1 as noted by the appearance of the absorption features of the photosynthetic complexes. **b** HPLC analysis of carotenoids extracted from *Rba. sphaeroides* strains. Heterologous expression of *crtI* from LLY-WYZ-10_1 (*Nannocystaceae*) in a mutant blocked at phytoene (Δ*crtIC*) demonstrates that this gene functions as phytoene desaturase synthesizing both neurosporene and lycopene, as judged from their absorption spectra. Source data are provided as a Source Data file.

reductase-encoding *bciA*[50] was selected as the host, to determine if the myxococcotal source of the genes has the ability to synthesize bacteriochlorophyll *a* or bacteriochlorophyll *b*. *Rba. sphaeroides*, along with all purple bacteria, assembles light-harvesting complex 1 around its RCII, forming the RC–LH1 supercomplex, and this organism also uses light-harvesting complex 2 (LH2) as a peripheral antenna[51]. These complexes determine the absorption profile of the organism, resulting in characteristic features in the whole cell absorption spectrum at 800 nm and 850 nm, contributed by LH2, and a feature appearing as a shoulder at 875 nm, contributed by RC–LH1 (Fig. 4a). The mutant lacking *bchYZ* and *bciA* cannot assemble these complexes, thus lacks these absorption bands, and displays altered pigmentation (Fig. 4a). It has been demonstrated that synthesis of BChl *b* in this mutant background does not restore light harvesting complex assembly[50]. We found that expression of *bchYZ* from a member of the *Ca*. Kuafubacteriaceae restored the pigmentation of these cells to that of the wild-type strain, and also restored assembly of LH2 and RC–LH1, as determined by the appearance of their relevant absorption features in the whole cell spectrum, confirming their function as an enzyme of bacteriochlorophyll *a* biosynthesis (Fig. 3). The catalytic components of COR, BchYZ, require electrons from reduced ferredoxin that are shuttled through the BchX protein[52]. In our complementation experiment, the native *Rba. sphaeroides* BchX protein must interact with the myxococcotal BchYZ to form an active chimeric enzyme. Despite the time since the horizontal transfer of a purple bacterial PGC to the *Myxococcota*, our results demonstrate that an active enzyme can be formed by proteins encoded in PGCs from extant bacteria from different phyla, indicating their well conversed evolutionary history.

*Myxococcota* are known as social organisms with a predatory or saprophytic lifestyle, and many cultured strains can form fruiting bodies to resist extreme environments[53], displaying flexible gene expression patterns that vary with environments[54]. Phototrophic capability may be an important feature for the survival of *Myxococcota* during intervals where prey and/or organic matter is limited. The metabolic model of aerobic anoxygenic phototrophs (AAPs) suggests that light can serve as an supplementary energy source to directly affect swimming speed of the marine proteobacterium *Dinoroseobacter shibae*[55]. In addition, it has

been reported that bacteriochlorophyll synthesis of *Roseobacter* sp. was induced under organic substrate limitation[56]. Meanwhile, photo-heterotrophic *D. shibae* was verified to use the additional energy from light to assimilate carbon dioxide for amino acid biosynthesis through the ethylmalonyl-CoA cycle (EMC, KEGG M00373)[57], and light was shown to enhance survival of *D. shibae* during long-term starvation[58]. The genomes of *Ca*. Houyibacterium oceanica, recovered from surface seawater, contain near complete EMC pathways with the key enzyme crotonyl-CoA carboxylase/reductase (encoded by the gene *ccr*, K14446). Intriguingly, these potentially phototrophic *Myxococcota* from the same families show significantly larger genome sizes compared with other non-phototrophic families within *Myxococcota* (Wilcoxon rank sum tests, $P < 0.001$; Fig. S7 and Supplementary Data 6). It might indicate that many additional genes are required for *Myxococcota* to perform phototrophic and other metabolic processes[31]. Therefore, incorporating phototrophic capability into genomes may supplement the energy requirement during motility, increase the production of secondary metabolites, and allow them to cope with survival challenges through flexibly regulated phototrophic lifestyle.

On the other hand, *Myxococcota* MAGs with phototrophic potential shared significantly higher GC content compared to either members with the same families or other members in this phylum (Wilcoxon rank sum tests, $P < 0.001$; Fig. S7). It is suggested that higher GC content is associated with aerobic lifestyle[59]. The conversion of Mg-protoporphyrin IX monomethyl ester to 3,8-divinyl-protochlorophyllide *a* is a key step in the production of (bacterio)chlorophylls in all chlorophototrophs[60] and is responsible for the characteristic green color associated with this family of pigments. This reaction is catalyzed by two distinct enzymes that use molecular oxygen (AcsF) or water (BchE) as oxygen donors, respectively. AcsF dominates in chlorophototrophs found in oxic environments (e.g., plants, *Cyanobacteria*, and AAPs), while BchE is essential for organisms performing photosynthesis in the absence of oxygen (e.g., *Firmicutes* and *Chlorobia*). MAGs from *Myxococcota* with PGC all encode an AcsF protein (Fig. S6), suggesting their phototrophy can operate with an aerobic lifestyle[59]. Intriguingly, all MAGs of *Ca*. Kuafubacteriaceae and some MAGs of *Polyangiaceae*, *Myxococcaceae*, and *Ca*. Houyibacteriaceae also contain *bchE* genes outside the PGCs,

indicating that some *Myxococcota* members are potential facultative anaerobes that perform phototrophy in anoxic conditions. Phylogenetic evidence (Fig. S6) indicates that BchE sequences of *Myxococcota* form a monophyletic branch clustering near the clades of *Gemmatimonadota* and *Proteobacteria* (average identities ~60–80%), suggesting that they are functional enzymes. On the contrary, ten MAGs of *Ca.* Houyibacterium oceanica recovered from surface seawater do not encode *bchE* genes, suggesting that anaerobic phototrophy is not a practical metabolic option in the oxygen-replete surface ocean and therefore was lost because of selective pressure. This is consistent with previous studies indicating that *Myxococcota* species are usually cultured or recovered from oxic environments[18].

Different genera from the phylum *Myxococcota* can be distinguished by their unique pigmentation, such as the orange color of the *Chondromyces* and the reddish to violet color of the *Archangium*[18]. *Myxococcota* with PGCs also contain diverse carotenoid biosynthesis pathways which may serve in light harvesting as accessory pigments and in photoprotection against harmful free radicals, or as antimicrobial agents and compounds with alternative functions. Besides the *crtCDEF* operon that is located in the PGC, multiple orthologs of carotenoid synthesis genes were dispersed throughout the chromosome. Some of them arranged occasionally as a carotenoid cluster, varying with different MAGs (Fig. S8). This complicated pattern suggests potential parallel acquisitions of carotenoid synthesis genes. Generally, all six families can synthesize spheroidene or spirilloxanthin (CrtF, K09846), but only *Ca.* Kuafubacteriaceae encode spheroidene monooxygenase (CrtA, K09847) to specifically recognize the C-2 position of spheroidene or the C-2 and C-2′ positions of spirilloxanthin to produce spheroidenone or di-ketospirilloxanthin[61]. Like many photosynthetic purple bacteria, the absence of CrtA is common in *Myxococcota*, while *crtA* genes are frequently located outside the PGC. Spheroidenone in *Ca.* Kuafubacteriaceae may serve as the main light harvesting carotenoid, as previously demonstrated in *Roseobacter*[62]. It was reported that spheroidenone is synthesized by anaerobic photoautotrophic organisms such as *Rhodobacter sphaeroides* and *Rhodovulum marinum* when grown under very low oxygen tension[63]. Besides, the potential to synthesize spirilloxanthin is consistent with the major carotenoids reported in *Congregibacter litoralis* KT71[64]. Except for *Nannocystaceae*, all families contain genes potentially coding for enzymes for synthesis of β-carotene (LcyB/CrtL1/CrtY, K06443; or CruA, K14605) and isorenieratene (CrtU/CruE, K09879) under suitable conditions. Additionally, *Ca.* Kuafubacteriaceae, *Myxococcaceae*, *Polyangiaceae*, and *Nannocystaceae* may further synthesize zeaxanthin (CrtR K02294; or CrtZ, K15746). These pigments may function in light harvesting, photoprotection, and sustaining membrane integrity[65]. Diverse carotenoids synthesized by *Ca.* Kuafubacteriaceae may play important roles in its physiology and ecology. Collectively, the substrate flexibility of many carotenoid biosynthesis enzymes makes it hard to predict the exact profile of pigments displayed by an organism based on its *crt*/*cru* gene complement. Therefore, further investigation will be necessary to determine the exact biosynthetic pathways employed by *Myxococcota*, and how they are regulated. Interestingly, proteorhodopsin (PR) was found in most MAGs of *Ca.* Houyibacterium oceanica belonging to the family *Ca.* Houyibacteriaceae, and a salt-lagoon-sourced MAG of *Ca.* Kuafuhalomonas phototrophica belonging to the family *Ca.* Kuafubacteriaceae (Supplementary Data 2). Meanwhile, the beta-carotene 15,15′-dioxygenase (Blh, K21817) can be identified in the genome of *Ca.* Kuafuhalomonas phototrophica but cannot be found in *Ca.* Houyibacterium oceanica. This phenomenon suggests that *Ca.* Kuafubacteriaceae and/or *Ca.* Houyibacteriaceae may have evolved the capability to complementarily perform both (BChl)-based and proton-pumping rhodopsin-based phototrophy in a single bacterium according to energy needs and environmental changes, as has recently been demonstrated for *Tardiphaga* sp.[66] and *Sphingomonas glacialis*[67].

The synthesis of spheroidene- or spirilloxanthin-type carotenoids is determined by the activity of the enzyme phytoene desaturase, encoded by *crtI*. Spheroidene-type carotenoids result from three desaturation/oxidation reactions on phytoene to produce neurosporene, while spirilloxanthin-type pigments are synthesized via four desaturations to form lycopene from phytoene[68]. To test whether *Nannocystaceae* are likely to produce spheroidene or spirilloxanthin-type carotenoids, the *crtI* gene was synthesized and cloned into a purple bacterial expression vector. The resulting plasmid was conjugated into a mutant of *Rba. sphaeroides* lacking *crtI* and *crtC*, thus is blocked at phytoene, and the product of any desaturation is prevented from further modification by native carotenoid biosynthesis enzymes. The Δ*crtC* mutant is unable to synthesize colored carotenoids (Fig. 4b). The transformed mutant was pigmented—carotenoids from this strain were extracted and analyzed by HPLC; two prominent peaks were observed in the chromatogram (Fig. 4b). Spectral analysis of these peaks demonstrated that the strain accumulated both neurosporene and lycopene (determined by the characteristic long-wavelength absorption maxima of these carotenoids, 470 and 505 nm, respectively). This result suggests that *Nannocystaceae* may display a complex profile of carotenoids, synthesizing a combination of spheroidene- and spirilloxanthin-type pigments as is observed in some purple phototrophs[69], but that the gene from this organism encodes a functional phytoene desaturase enzyme.

## Autotrophy and additional metabolic modes

Up until now, all cultured strains of *Myxococcota* are heterotrophic. The complete glycolysis (Embden-Meyerhof pathway, KEGG M00001) and citrate cycles (TCA cycle/Krebs cycle, KEGG M00009) are present in all genomes here, indicating heterotrophic lifestyle of members in these *Myxococcota* with PGC. We further checked carbon fixation pathways adopted by previously studied phototrophs, including the Calvin-Benson-Bassham (CBB) cycle (KEGG pathway M00165), reductive citrate cycle (rTCA cycle, KEGG M00173), 3-hydroxypropionate bicycle (3HP, KEGG M00376), for these *Myxococcota* containing photosynthesis genes. Astonishingly, we found some members of them may conduct autotrophic metabolism using the CBB cycle. Complete or near complete CBB cycles were identified in the families *Nannocystaceae*, *Ca.* Kuafubacteriaceae, and *Myxococcaceae*. Specifically, the genome of *Ca.* Xihebacterium glacialis from the family *Nannocystaceae*, recovered from glacier surface soil, contains a complete CBB cycle. Meanwhile, the genome of *Ca.* Xihepedomonas phototrophica of the family *Myxococcaceae*, recovered from Arctic peat soil, and genomes of *Ca.* Kuafubacterium phototrophica and *Ca.* Kuafucaenimonas phototrophica of the family *Ca.* Kuafubacteriaceae, also contain near complete CBB cycles with genes encoding the key enzymes ribulose-1,5-bisphosphate carboxylase/oxygenase (RuBisCO) and phosphoribulokinase (PRK). Phylogenies place the RuBisCO sequences of aforementioned species into form I RuBisCO (Fig. S9), which is generally found in photosynthetic algae, plants, and bacteria[70]. The RuBisCO sequences of these families have average identities ~80–90% to the known protein sequences in the NCBI nr database, and they were clustered as three different subclades in the form I RuBisCO tree (Fig. S10), indicating that CBB cycle-based autotrophy may come from multiple historical HGT events, e.g., from *Proteobacteria*.

For other potential carbon, sulfur and nitrogen metabolic pathways, we found genomes of *Ca.* Kuafubacteriaceae and *Myxococcaceae* contain complete phosphate acetyltransferase-acetate kinase pathways (KEGG M00579). Furthermore, all genomes also encode complete assimilatory sulfate reduction (ASR, KEGG M00176), except for the genomes from the families *Myxococcaceae* and *Nannocystaceae*, which lack genes coding for phosphoadenosine phosphosulfate reductase (CysH). To our knowledge, sulfur respiration in *Myxococcota* has not been verified, but similar features have also been described in the MAGs assembled from sulfur and sulfide-rich Zodletone Spring[71]. *Ca.* Kuafubacteriaceae, *Myxococcaceae*, *Polyangiaceae* also have the

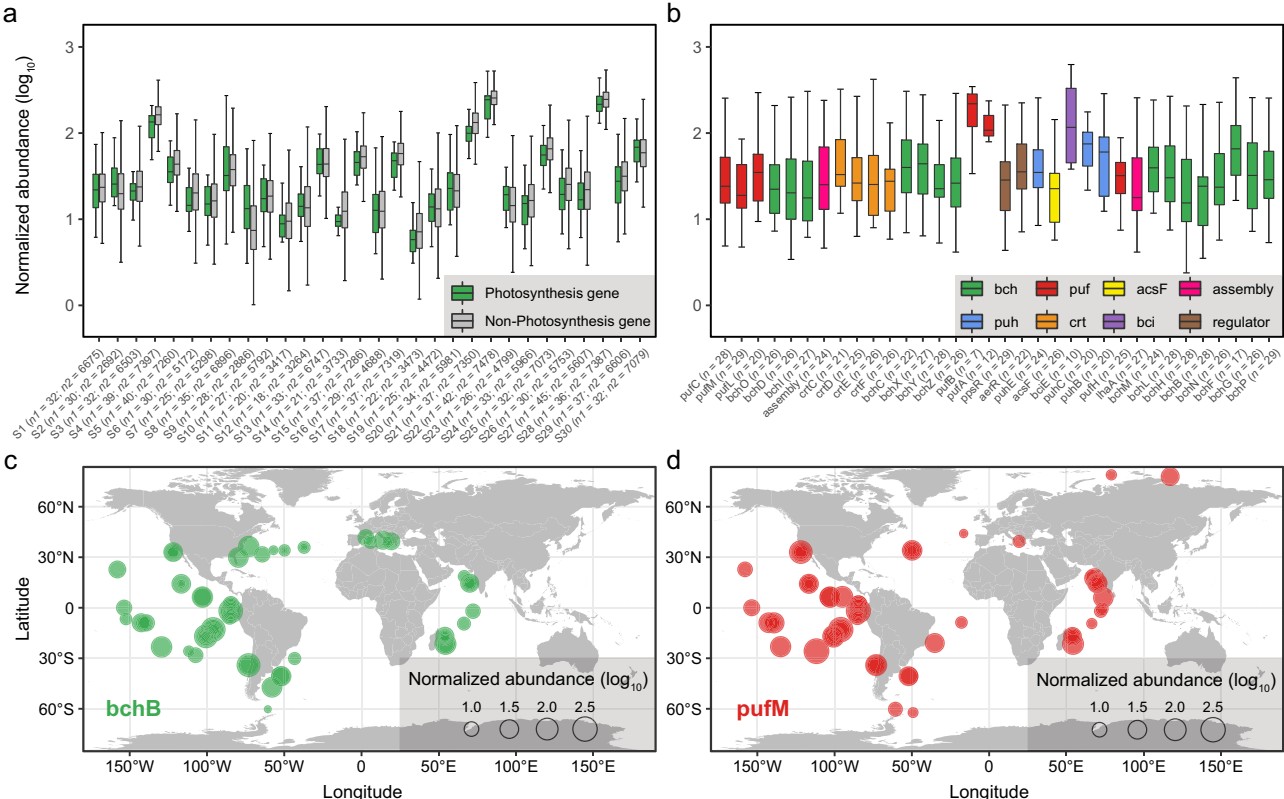

**Fig. 5 | The expression pattern of photosynthesis genes of *Myxococcota* on a global scale. a** The expression level (FPKB, fragments per kilobase per billion metatranscriptomic fragments) of photosynthesis genes (green) compared with non-photosynthesis genes (gray) of six representative *Myxococcota* genomes in typical samples with photosynthesis gene prevalence >50% and mean FPKB > 3. *n1* values refer to the number of independent results of photosynthesis genes. *n2* values refer to the number of independent results of non-photosynthesis genes (Detailed information of the name and values are listed in the Source Data file).

**b** The expression of PGC of representative genome (LLY-WYZ-15_3) from the family *Ca*. Houyibacteriaceae in typical samples. The gene order in the boxplot is identical to the PGC of *Ca*. Houyibacteriaceae as shown in the Fig. 2c. *n* values refer to the number of independent samples. **c, d** The maps show the global expression pattern of *Myxococcota* photosynthesis genes *bchB* and *pufM*, respectively. The colors of photosynthesis genes are identical to the Fig. 2c. Box plots indicate median (middle line), 25th, 75th percentile (box) and 1.5 times interquartile range (whiskers), and outliers are not shown. Source data are provided as a Source Data file.

potential for dissimilatory nitrate reduction (DNR, KEGG M00530). These additional features may complement the metabolic potentials of versatile *Myxococcota*, which have previously been reported including denitrification, dehalorespiration, and facultative anaerobic respiration[72].

Together, multiple lineages of the phylum *Myxococcota* have the potential to perform phototrophy, which may be either photo-autotrophic or photoheterotrophic lifestyles depending on their survival pressure, such as in extremely oligotrophic environments versus eutrophic environments. Consequently, *Myxococcota* with photosynthesis genes evolved diversified metabolic potentials. With the annotation predictions from their genomes, (i) *Ca*. Kuafubacteriaceae should be a versatile generalist with the ability to perform anoxygenic photosynthesis coupled to CBB cycle-based carbon fixation, possessing the potential to synthesize a broad range of carotenoids (e.g., astaxanthin, spheroidenone, isorenieratene and spirilloxanthin), as well as ASR and DNR capabilities, and display a facultative anaerobic lifestyle. (ii) Members of *Myxococcaceae* and *Polyangiaceae* share many metabolic features in common, such as synthesizing specific carotenoids (e.g., zeaxanthin, spheroidene, isorenieratene and spirilloxanthin), performing ASR or DNR, and the ability to survive either aerobically or as facultative anaerobes. (iii) *Nannocystaceae* is characterized by aerobic predation incorporating anoxygenic photosynthesis coupled to CBB cycle-based carbon fixation, carotenoid synthesis (e.g., spheroidene and spirilloxanthin), and partial ASR. The photosynthesis of aerobic *Nannocystaceae* expands our current understandings of AAPs, which were considered to be limited to

photoheterotrophy. (iv) *Sandaracinaceae* may be aerobic photo-heterotrophic predators with the ability to perform ASR and synthesize several carotenoids (e.g., spheroidene, isorenieratene and spirilloxanthin). (v) Phototrophic *Ca*. Houyibacteriaceae differentiated into two species, aerobic *Ca*. Houyibacterium oceanica that contain the EMC pathway and PR, and the facultative anaerobic *Ca*. Houyihalomonas phototrophica that lack the EMC pathway and PR. These two species share similar potential for chlorophototrophy, carotenoid synthesis and ASR with the family *Sandaracinaceae*.

## Ecological and evolutionary implications

The discovery of potential photosynthetic capacity in *Myxococcota* which are known for their versatile lifestyle such as social and predatory behaviors, is intriguing. Despite studies finding that pigments play a role in mediating light-related functions, e.g., carotenoids function as photoreceptors in model organism *M. xanthus*[73], photosynthetic genes (e.g., *pufLM*) have previously not been found in any species of *Myxococcota*. In this study, potential phototrophic members of *Myxococcota* were recovered from diverse habitats spanning large-scale geographical distance (Fig. 2a, b), including aquatic and terrestrial habitats, salt lagoon and freshwater lake, oligotrophic and eutrophic seawater, high- and low-latitude geographical zones, as well as anthropogenic ecosystems. Furthermore, we dug into global ocean metatranscriptomic analyses to investigate whether their phototrophic capacity is active in situ. We found that photosynthesis genes of *Myxococcota* were actively expressed on a global scale, e.g., substantial expression of PGC in both coastal and oceanic zones across

tropical and subtropical oceans (Fig. 5). Predatory *Myxococcota* have already been shown to be keystone taxa in microbial food webs, shaping microbial community structure[74]. Consequently, the presence and expression of photosynthetic genes point to the possibility that *Myxococcota* are not only consumers shaping microbial communities by means of top-down control[75], but also may be primary producers. Together with the widespread occurrence of *Myxococcota* in natural ecosystems (Supplementary Data 7), this taxon with multiple ecological roles should contribute substantially to global biogeochemical cycles. Moreover, the cosmopolitan distribution of myxococcotal descendants suggests that versatile metabolisms enable them survive across various habitats, and their multifunctionality expands our understanding of microbial evolution and ecology.

Considering that phototrophic potential occurs in at least four clades (i.e., the families *Myxococcaceae*, *Polyangiaceae*, *Nannocystaceae*, and *Sandaracinaceae*) that have many verified predatory members (Figs. S2 and S3), we suggest two assumptions to explain the origin of phototrophic capability of *Myxococcota*: (i) "Transitional State" assumption states that the ancestors of *Myxococcota* and/or *Proteobacteria* may have pioneered the invention of anoxygenic phototrophy in the late Archaean Earth when oxygen was accumulating, whereas the predatory capacity evolved later in *Myxococcota*; (ii) "Pillage and Plunder" (assimilating prey genes) assumption explains that if myxococcotal ancestors are predatory, frequent predatory behavior would provide more opportunities for them to horizontally gain foreign genes (e.g., PGC). It is uncertain whether the crown group of phototrophic members is simultaneously predatory (i.e., chimeric photosynthetic predators) or whether photosynthetic and predatory capacities represent two independent metabolic processes of different species (evolutionary adaptions to environmental pressure). In addition, current in silico analyses cannot directly prove that *Myxococcota* with PGCs actually perform photosynthesis in vivo. Consequently, cultivation of representatives of these *Myxococcota* will be required to validate the photosynthetic activity and hypotheses suggested in this study as well as other important capacities and bioactive resources in myxococcotal biology. Most importantly, although these implications are intriguing, there are still many mysteries to be explored within this fascinating lineage.

In conclusion, photosynthesis fuels the surface ecosystems on Earth. However, microbes employing bacteriochlorophyll-based reaction centers have been found in only eight out of the 148 currently-described bacterial phyla. According to a global-scale comparative genome analysis for phototrophic diversity, we obtained 32 genomes with chlorophototrophic genes belonging to a previously unrecognized predatory bacterial phylum, *Myxococcota*. Phototrophic members of this phylum possess and actively express bacteriochlorophyll biosynthesis and reaction center genes in distinct ecosystems across the globe. Heterologous expression of the pigment biosynthesis genes further suggest their ability to grow as phototrophs. Phototrophy in *Myxococcota* is not restricted to a single lineage, but occurs in at least six discrete families, interspersed with non-phototrophic members. Remarkably, *Nannocystaceae*, *Ca.* Kuafubacteriaceae, and *Myxococcaceae* encode (near) complete Calvin-Benson-Bassham (CBB) cycle with the form I ribulose-1,5-bisphosphate carboxylase/oxygenase (RuBisCO). The closely related phylogenies of photosynthesis genes from multiple lineages of *Myxococcota* suggest that phototrophy might be an ancient trait, with subsequent independent losses of phototrophic ability in many lineages. Overall, the discovery of anoxygenic phototrophy in *Myxococcota* extends the known distribution of phototrophic diversity in the *Bacteria* domain, and reveals overlooked ecological roles of *Myxococcota* in diverse ecosystems.

## Methods
### Data collection and treatments
Genomes used for mining potential phototrophs include two parts, MAGs binned from assembled metagenomic sequences and genomes downloaded from public databases. Metagenomic datasets were downloaded from the NCBI Sequence Read Archive public database (https://www.ncbi.nlm.nih.gov/sra). The Sickle algorithm version 1.33 (https://github.com/najoshi/sickle) was used to trim metagenomic sequencing reads. The trimmed reads were subsequently assembled by MEGAHIT[76] version 1.2.9 with k-min 27 and k-max 127. The sequence coverage of contigs was calculated by mapping the trimmed reads to each contig using BBMap version 38.18[77]. The assembled metagenomic sequences were binned using MetaBAT[78] version 2:2.15 with 1.5 kb as contig length cut-offs. Additionally, a total of 474,091 genomes of prokaryotes were downloaded from the NCBI prokaryotes database (https://ftp.ncbi.nlm.nih.gov/genomes/genbank; 421,576; July 2022) as well as the GEM catalog[79] (https://genome.jgi.doe.gov/portal/GEMs/GEMs.home.html; 52,515; February 2022), respectively. The metadata of *Myxococcota* genomes was determined manually by searching GEM metadata, NCBI Datasets (https://www.ncbi.nlm.nih.gov/data-hub/genome), NCBI Biosample (https://www.ncbi.nlm.nih.gov/biosample), and scientific literature.

### Photosynthesis gene identification
All genomes were subjected to open reading frame (ORF) prediction with Prodigal[80] version 2.6.3 with parameter -p single. A local database of representative sequences including bacteriochlorophyll and carotenoid biosynthesis, RC, electron transport, antenna systems, regulatory proteins, and cofactors, was collected from published literatures and KEGG database, and subsequently manually curated. The built local database was used as subject, and the predicted ORFs were queried against the database using DIAMOND[81] version 2.0.14.152 with parameters coverage >75% and $e < 1 \times 10^{-20}$. Genomes containing a partial to complete (bacterio)chlorophyll biosynthesis pathways were screened as potential chlorophototrophs. Taxonomic assignment was classified using GTDB-Tk version 2.0.0[35] and by placement in a concatenated ribosomal protein phylogeny[82,83]. A total of 8,221 genomes containing >50% of the genes of (bacterio)chlorophyll biosynthesis pathway were retained. These genomes were also screened for reaction center proteins, including the iron-sulfur containing RCI type utilized by green sulfur bacteria (the class *Chlorobia* of the phylum *Bacteroidota*), *Heliobacteria* (*Firmicutes*), and *Chloracidobacteria* (*Acidobacteriota*), the pheophytin-quinone RCII type employed by purple bacteria (*Proteobacteria*), green non-sulfur bacteria/filamentous anoxygenic phototrophs (FAP) (*Chloroflexota*), *Gemmatimonadota*, and *Eremiobacterota*, and photosystem I (PSI) and photosystem II (PSII) harbored by *Cyanobacteria*[3,5–12,84].

To determine the phylogenies of concatenated PGCs, we first empirically set two type thresholds to locate the PGCs from all potential chlorophototrophs. A rigorous threshold is defined as at least 25 photosynthesis related genes should distribute on a fragment with no more than 50 continuous genes. Similarly, a moderate threshold represents that at least 12 photosynthesis genes should distribute on a fragment with no more than 24 continuous genes. The rigorous threshold is expected to determine all complete or nearly complete PGCs, while the moderate threshold should be able to ensure that fragmented PGCs are recovered even if the PGCs are incomplete or the supposed complete PGCs distribute in distinct genomic fragments. With the rigorous threshold, (near) complete PGCs and corresponding genomes were identified from the BLASTP results based on the software DIAMOND[81]. A total of 1,845 PGCs, with gene number ranging between 25 to 41, were retained from the 8,221 potential phototrophic genomes. Subsequently, all phototrophic genomes with PGCs were dereplicated using dRep[85] version 3.4.0 with parameter -sa 0.95 to reduce redundancy and improve accuracy. PGC proteins occurring in at least 70% of non-redundant genomes were selected as PGC markers (Supplementary Data 8). The sequences of all PGC marker proteins and genes were retrieved from the non-redundant phototrophic genomes.

## Evaluations and comparisons of genomes

Potential chlorophototrophic genomes were evaluated by CheckM[86] version 1.1.3 using lineage-specific marker genes with parameter lineage_wf, and reference genomes with completeness >70% and contamination <10% were considered for further analyses. The approximate genome size was estimated by the following formula: approximate genome size equals assembly genome size/estimated completeness/(1+estimated contamination). The Wilcoxon rank-sum test was applied to compare the approximate genome size and GC content between phototrophic and nonphototrophic genomes using the wilcox.test function in the "stats" package[87]. The completeness and contamination of *Myxococcota* MAG were also recertified by machine learning based CheckM2[36]. To check the protein identities of *Myxococcota* to the most closely related sequences, the predicted ORFs were also queried against the NCBI nr protein database (August 2022) and eggnog[88] database with the BLASTP algorithm (coverage > 75% and e < 1 × 10$^{-20}$) using DIAMOND version 2.0.14.152 (Supplementary Data 9). Genomes were assessed for AAIs using CompareM version v0.1.2 with parameter aai_wf (https://github.com/dparks1134/CompareM). ANIs were estimated using OrthoANIu (v.1.2)[89]. CoverM (https://github.com/wwood/CoverM) was subsequently used to calculate the genome relative abundance based on coverage estimation. The web portal GhostKOALA on the KEGG website[90] was employed for metabolic pathway analyses.

## Phylogenetic analyses based on sequences of conserved proteins, PGCs, and PufLM/BchHDI/BchLNB/AcsF/BchE/RuBisCO

For phylogenomic analysis, 232 representative bacterial genomes were screened from the previously downloaded NCBI prokaryote genome database (https://ftp.ncbi.nlm.nih.gov/genomes/genbank). Based on a concatenated alignment of a set of 37 marker proteins, all MAGs of *Myxococcota* with photosynthesis genes and the reference genomes were used to construct a phylogenomic tree. The protein sequences of PGCs and single-protein markers (PufLM, BchHDI, BchLNB, AcsF, and BchE) were also retrieved from the genomes. The reference protein sequences of RuBisCO were retrieved from the published paper[70]. Protein sequences were aligned by the MAFFT[91] algorithm version 7.490 with parameters --genafpair –ep 0 --maxiterate 1,000. The alignment was then filtered using trimAl[92] version 1.4.1 with parameter -automated1. Subsequently, the trimmed 37 markers and PGCs were respectively concatenated into a single alignment, and were used to construct the phylogenetic trees by employing IQ-Tree[93] and FastTree[93], respectively. Divergence timing analyses were estimated based on the phylogenomic tree using the treePL[94], and Reltime method[95] as implemented in MEGA11[96]. Three different age constraints were used, including the potential fossil evidences of the cyanobacterial clades *Nostocales* and *Stigonematales* (-1.7 Ga), the predicted crown group of oxygenic *Cyanobacteria* (2.42-2.97 Ga), and the inferred bacterial root lineages (-3.8 Ga)[97,98]. Furthermore, the 32 MAGs and all high-quality *Myxococcota* genomes available from the GTDB r207 were used to construct phylogenetic trees based on the concatenated alignments of 120 conserved single-copy marker genes using GTDB-Tk[35]. Detailed descriptions of parameters of phylogenetic trees are provided in Supplementary Data 10.

## Transcriptomic profiling of potential phototrophic *Myxococcota*

Metatranscriptomic datasets from *Tara* Oceans projects[99–101] were downloaded from the NCBI Sequence Read Archive public database (https://www.ncbi.nlm.nih.gov/sra). The fastp algorithm version 0.12.4[102] was used to trim raw metatranscriptomic sequencing reads. For the quantification of transcript expression, we used salmon version 1.9.0[103] to map the trimmed metatranscriptomic reads to the six representative *Myxococcota* genomes. The abundance of each gene in metatranscriptomes was calculated as the metric-FPKB (fragments per kilobase per billion metatranscriptomic fragments) normalized based on the gene length and sequencing depth. For the profile of global expression of *Myxococcota* photosynthesis genes, we summed the FPKB of genes that belongs to the same function to calculate the functional abundance of each metatranscriptomic library.

## Experimental validation for key enzymes involved in bacteriochlorophyll and carotenoid biosynthesis

**Growth of described strains.** *Rba. sphaeroides* strains were grown at 30 °C under microoxic conditions in the dark in an orbital shaker at 180 rpm, or phototrophically in anoxic sodium succinate medium 27 (N27 medium)[104] under illumination (20 μmol·photons·m$^{-2}$·s$^{-1}$) provided by 60 W halogen bulbs. All phototrophic culture transfers were performed in an anoxic chamber (Coy Laboratories). When carrying expression plasmids, medium was supplemented with kanamycin at 30 μg·ml$^{-1}$. *E. coli* strains JM109[105] and ST18 (DSM 22074)[106] transformed with pBBRBB-Ppuf$_{843-1200}$ plasmids[107] were grown in a rotary shaker at 37 °C in lysogeny broth supplemented with 30 μg·ml$^{-1}$ kanamycin. ST18 cells were supplemented with 50 μg·ml$^{-1}$ 5-aminolevulinic acid (ALA).

**Complementation of *Rba. sphaeroides* mutants in trans.** The exact sequences of *Ca.* Kuafubacteriaceae *bchYZ* and *Nannocystaceae crtI* could not be commercially-synthesized, nor could versions codon-optimized for expression in *Rba. sphaeroides*, due to their extreme GC-content. To overcome this, a codon optimization tool (www.idtdna.com/codon/optimization) was used to generate sequences optimized for expression in *E. coli*, codons used by *Rba. sphaeroides* at low frequency were manually changed, and restriction sites for subsequent cloning were also removed (sequences can be found in Supplementary Note 2). The resulting genes were synthesized flanked by BglII and SpeI restriction sites, these enzymes were used to excise these fragments, which were cloned in place of DsRed-Express2 in pBBRBB-Ppuf$_{843-1200}$[107]. The resulting plasmids isolated from JM109 were verified by DNA sequencing and conjugated into *Rba. sphaeroides* mutants from the ST18 strain of *E. coli*. Transconjugants harboring the plasmids were selected on N27 medium supplemented with kanamycin.

**Absorption spectroscopy.** Ultraviolet/visible/near-infrared absorption spectra were collected on a Cary 3500 spectrophotometer (Agilent Technologies) scanning between 300 and 1100 nm at 1 nm intervals with a 0.1 s integration time.

**Pigment extraction.** *Rba. sphaeroides* cultures were washed in 50 mM Tris-HCl pH 8.0 and pelleted by centrifugation. Total pigments were extracted, concentrated and reconstituted according to Namoon, et al.[108].

**Pigment analysis.** Extracted carotenoids were separated on an Agilent 1100 HPLC system equipped with a Supelco Discovery HS C18 column (5 μm particle size, 120 Å pore size, 250 × 4.6 mm) maintained at 40 °C using a program modified from Magdaong, et al.[109]. Pigments were eluted on a 50 min isocratic gradient of 58:35:7 (v/v/v) acetonitrile/methanol/tetrahydrofuran. Elution of neurosporene and lycopene were monitored at 470 nm and 505 nm, respectively.

## Automated nomenclature

The method Great Automatic Nomenclator (GAN)[110] was employed to create linguistically correct taxonomic names for *Myxococcota* MAGs. Fourteen genus names were proposed for these uncultured *Myxococcota* (Supplementary Data 11) by combinatorial concatenation of three types of word roots (Chinese mythological figures correlated with the sun, the habitat terms, and universal Latin endings or diminutives, respectively) (Supplementary Data 12) and considering International Code of Nomenclature of Prokaryotes (ICNP) rules

governing the use or elision of connecting vowels. The type materials for each proposed genus are designated as those with satisfied genome quality criterion[111] (or the MAG with highest quality when MAGs do not meet the criteria) and environmental representativity. Higher-level taxonomic ranks were generated by concatenating the stem with an appropriate suffix according to the ICNP rules[112].

## Selection of representative MAGs for visualization

Representative genomes of photosynthesis gene-containing *Myxococcota* were selected for the data visualization of PGC arrangement, metabolic schemes, and expression pattern of photosynthesis genes. The MAGs were dereplicated using dRep[85] version 3.4.0 with parameter -sa 0.95. Representative myxococcotal genomes were identified based on the genome quality, PGC quality, and completeness of metabolic pathways. In detail, for the visualization of PGC arrangement and expression pattern of photosynthesis genes, all PGCs were firstly compared, and representative MAGs with complete PGC and high genome quality were selected from each family. For the metabolic schemes, MAGs with high genome quality and complete metabolic pathways are labeled as representatives, and other MAGs that have additional genes to supplement the representatives are labeled as "Additional" genomes. We also selected the MAG LLY-WYZ-15_3 that was recovered from ocean as representative of *Ca.* Houyibacteriaceae to visualize PGC expressions for samples from *Tara* Oceans. The information of representative MAGs is listed in Supplementary Data 2.

## Reporting summary

Further information on research design is available in the Nature Portfolio Reporting Summary linked to this article.

## Data availability

The genomes of *Myxococcota* with photosynthesis genes generated in this study have been deposited in the NCBI GenBank under BioProject ID PRJNA943119, as well as the eLibrary of Microbial Systematics and Genomics (eLMSG; https://www.biosino.org/elmsg/index) under accession numbers LMSG_G000011443.1, LMSG_G000011444.1, LMSG_G000011445.1, LMSG_G000011446.1, LMSG_G000011447.1, LMSG_G000011448.1, LMSG_G000011449.1, LMSG_G000011450.1, LMSG_G000011451.1, LMSG_G000011452.1. The databases used in this study include NCBI database (https://www.ncbi.nlm.nih.gov/), GEM catalog (https://genome.jgi.doe.gov/portal/GEMs/GEMs.home.html), GTDB database Release 207 (https://data.gtdb.ecogenomic.org/releases/release207/), and KEGG database (https://www.kegg.jp/kegg/). Source data are provided with this paper.

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

## Acknowledgements

This work is supported by the National Natural Science Foundation of China (NSFC grant No. 92051116, 92251303, 91951209, 42272354) and the Natural Science Foundation of Shanghai (20ZR1428000). N.M.R. is supported by a Doctoral Training Partnership from the Biotechnology and Biological Sciences Research Council (BBSRC). D.P.C. acknowledges support from BBSRC grant BB/W008076/1. We are grateful to the researchers who shared their sequence data on the NCBI (https://www.ncbi.nlm.nih.gov/) and to the US Department of Energy Joint Genome Institute (https://www.jgi.doe.gov/) for providing genome files in collaboration with the scientific community. We thank Prof. Guoqing Zhang, Ruifang Cao, Wan Liu for their help on data preparation and submission. This work is also supported by the MASH-Ocean consortium. Part of our computations were run on the Siyuan-1 cluster supported by the Center for High Performance Computing at Shanghai Jiao Tong University (SJTU HPC). We are also thankful to Chen Li, Yujue Wang, and Wu Guo from SJTU HPC for useful discussion.

## Author contributions

L.L. and Y.W. designed the research, performed the analyses, and wrote the paper. L.L., D.H. and Y.H. collected data. N.M.R. performed the heterologous expression experiments and their analyses. D.P.C. provided knowledge on metabolism of phototrophs and wrote the paper. F.W. provided guidance and wrote the paper.

## Competing interests

The authors declare no competing interests.
