## [Peer Review File - NEW · Nature Communications]

Globally distributed Myxococcota with photosynthesis gene clusters illuminate the origin and evolution of a potentially chimeric lifestyleEditorial Note: Parts of this Peer Review File have been redacted as indicated to remove third-party material where no permission to publish could be obtained.

Reviewer #1 (Remarks to the Author):

In this submission titled "Globally distributed phototrophic prokaryotic predators illuminate the origin and evolution of a novel chimeric lifestyle", Li and colleagues report their original finding that a diverse group of myxobacteria (recently classified as Myxococcota) encodes whole gene clusters known to enable photosynthetic activity. Given that several myxobacteria are known predators, they set out to demonstrate that the parallel presence of homologous genes tied to photosynthesis and heterotrophic predation mean that these species engage in chimeric life styles. As such, this finding would be highly intriguing and of high interest to a larger scientific community, especially for scientists in microbiology and botany. Yet, there are several major caveats that I will highlight further down below, which prevents this contribution to substantiate the claims it makes.

In more detail, the authors resorted to genome mining techniques to identify the presence of photosynthesis-related genes in metagenomes. They de novo assembled metagenomic reads (that are dubbed Metagenome-Assembled Genomes, or MAGs) and queried assemblies against a database containing an exhaustive list of relevant bacterial photosynthesis genes with parameters chosen to assure reasonable best matches (they also added a manual curation step of search results). Within the set of identified MAGs the authors quite surprisingly identified conserved photosynthesis-related gene clusters within the class of Myxococcota, including in several known families, but also in two entirely novel groups (to which they assign the novel names, Kuafubacteria, and Houyibacteriaceae). Given this observation, the authors would like to draw a strong connection between phototrophy and heterotrophic predation, the latter off which is a feature of many culturable myxobacteria, including *Myxococcus xanthus*.

The way they approach this, however, is shaky, and as such the connection between photosynthesis and predation remains highly speculative. Below I substantiate my concerns and give a verdict.

The good: identification of photosynthesis-related gene clusters in myxobacteria

There is no question in my mind that what the paper found is completely new: the identification of photosynthesis-related gene clusters in MAGs, taxonomically classifiable as Myxococcota, of which several representatives are model systems for multicellular development, sociality, and predation. I really like their phylogenetic approaches, these meet accepted standards, and their graphical representations are clear and aesthetically appealing. They also present phylogenetic evidence and gene expression data to further demonstrate that many or most of the complete sets of photosynthesis genes are conserved and expressed in nature. This approach seems very robust to me, and demonstrates that photosynthesis may be important in several myxobacterial clades.

The bad (I): the interpretation of active photosynthesis from expression data

No myxobacterial research group to date (including the great Dale Kaiser) has previously found evidence for photosynthesis in myxobacteria. As such, the presence of these gene clusters and even the knowledge of gene expression is surprising and very exciting. But in my mind, in itself it is not direct proof that myxobacteria actually perform photosynthesis (however likely this appears to be). It has been shown in molecular detail that *Myxococcus xanthus*, the most commonly used model system in the field, is able to respond to light (via photoreceptors) by producing pigments. This line of research has been conducted mainly by the group of Montserrat Elías-Arnanz (see for example: <https://doi.org/10.1126/science.aay1436>). I think it remains to be seen if the photosynthesis-related genes have maintained their original purpose in myxobacteria (over all the billions of years), or if they were repurposed for some other tasks. MAGs alone won't tell us.

The bad (II): the conceptualisation of and lines of evidence for predation

The way the authors want to tie potential photosynthesis to potential predation is rather weak.

This starts with the fact that it remains elusive what the authors think bacterial predation is, and what would be accepted hallmark genes involved in bacterial predation, especially for myxobacteria. Myxobacteria form multicellular groups in order to overwhelm their prey, but can also kill by contact-dependance. While predatory life-styles are wide-spread across the myxobacteria, it is not yet clear if myxobacteria are generally predatory (see ref. 30 which the authors cited). In fact, many are saprophytes and also produce antimicrobials to fend off competitors, e.g., the cellulose-degrader *Sorangium cellulosum*, which the authors erroneously list as facultative predator.

While several killing mechanisms have been elucidated (eg, <https://doi.org/10.7554/eLife.72409>), just killing others is not predation. And, the general molecular basis of predation in myxobacteria is essentially still a mystery and the quest for elucidation of molecular elements has only recently begun.

To counter this knowledge gap, the authors resort to a machine-learning approach to infer common "predatory" genes. It is not entirely clear how this works to me, and so I am sceptical of the reported results., and this is mostly because of the list of genes it produces (see below).

These they report mainly in Tables S5 and S6, where they list what the authors define as "predators" (Table S5) and "predator-specific KEGG protein groups" (Table S6), respectively.

But not all members of Table S5 are predators, in fact *Sorangium cellulosum* is cellulolytic, as mentioned above, and can produces antimicrobial compounds meant for competitor control (killing) but not directly to use them as food source (they specialize on cellulose). Hence, any shared genes might not reflect predation per se, but simply antagonistic behaviors.

It is then not surprising, that Table S6 lists a lot of curious genes, which the authors call predation genes. Among them (on top) is listed *lacI*, the well characterized repressor of expression of genes encoding lactose degradation enzymes *lacZ*, *lacY* and *lacA*. *LacI* is a gene that is also present and perhaps better understood in *E. coli*, not known to be predatory. The list also includes dozens of seemingly unspecific genes including ammonium transporters. To me it remains unclear what these genes represent, and how they can be labelled predation genes. And as such, I think this analysis is not conclusive.

The ugly: ignoring well-established myxobacterial biology (which undermines their interpretation)

The most crucial issue arises from a lack of understanding of myxobacterial biology, which undermines the interpretation of the author's findings. Myxobacterial biology centers around two main pillars: multicellular development and social motility, both of which are clearly much better understood in molecular detail than predation is. Instead of using flagella to swim, most well-studied myxobacteria utilize two very different motility systems for movement across surfaces: social pili-driven motility (<https://doi.org/10.1038/s41467-020-18803-z>), and adventurous gliding motility (<https://doi.org/10.1073/pnas.1219982110>). Yet, in Figure 3 the authors depict the typical myxobacterium with a single flagellum not resembling anything that is usually studied in the myxobacterial community. What is more, the standard myxobacterial response to starvation is fruiting body formation in which the cells engage in highly coordinated multicellular development (and sticking to a precise developmental program). This part of the life cycle is extremely well studied, yet the authors only speculate that: "The phototrophic capability may be an important feature for the survival of predatory Myxococcota during intervals where prey is limited." (LL. 185 – 186). In other words, they do not apply the more standard knowledge on development, but rather resort to a fictional myxobacterial model that fits their data best.

The verdict:

In essence, the authors find something quite amazing, and I think then overreach by setting out to establish a link between the clear presence of photosynthesis-related genes in myxobacteria and the possibility of a chimeric life-style involving heterotrophic predation. In that respect they fail, so I think it is best for them to thoroughly reconsider the scope of their study, such that they can embed their incredibly interesting finding in a bullet-proof context. For that they need to stick to what can be reasonably shown using omics techniques alone, and incorporate well-established

facts about myxobacterial biology that were gathered in the past around seventy years. The latter research has not found any evidence for photosynthesis in myxobacteria.

Specific comments:

- The need to tone down any bold statement that remain unsubstantiated, (e.g., LL. 113-114: for the phototrophic Myxococcota)
- The very book chapter the authors cite as source that claims that all myxobacteria are predatory (ref. 30) states the following "It is not yet known whether all myxobacteria are predatory, however it seems likely that it is a common feature of the order; predators have been isolated from all three myxobacterial sub-orders, including representatives from the Chondromyces, Coralloccoccus, Enhygromyxa, Myxococcus, Plesiocystis, Pyxidicoccus, Racemicystis, Sorangium and Stigmatella genera (Amiri Moghaddam et al. 2016, 2018; Awal et al. 2016; Livingstone et al. 2017; Perez et al. 2016)." So, general predation is not verified truth but rather needs to be properly tested.
- LL. 72-73 missing information what is meant by «near» completeness of (bacterio)chlorophyll biosynthesis pathways.
- L. 131 "The highly similar topologies of photosynthesis gene trees". To make that statement, did the authors perform a statistical test to test for (in)congruence of the tree topologies (eg, using a Kishino–Hasegawa test)?
- Figure 1:
 - o add color information to those taxa that are non-phototrophic (I think the color used is a shade of brown).
 - o better highlight all taxa that belong to the myxobacteria in the trees (ie, by a clearer line of demarkation and circle sector enclosing all Myxococcota).

Reviewer #2 (Remarks to the Author):

The study by Li et al reports the first discovery of chlorophototrophic members of the predatory bacterial phylum Myxococcota and expands the number of phototrophic bacterial phyla to nine. They show the photosynthetic genes are from a monophyletic origin and also show vertical evolution. The discovery represents the first time a chimeric lifestyle of photosynthesis and predation are integrated into a single prokaryotic cell.

They assembled genomes from the NCBI prokaryotic database and the Earth's Microbiome catalog, which was used as queries for screening photosynthesis related homologs resulting in the discovery of the Myxococcota MAGs. They created gene trees from the photosynthetic genes to compare them with genes from other phyla and inferred metabolic schemes based on the presence of genes. They also use metatranscriptional data to determine the expression of photosynthesis genes for the different samples and propose the potential chimeric lifestyle of Myxococcota which involves photosynthesis, heterotrophy and predation.

The manuscript reads well and I think the work will be a significant addition to the field, especially in regards to the evolution of photosynthesis. The work supports the conclusions and claims. Below I have listed comments to address. The methods are sound and the work meets the expected standards in the field.

Comments for the main text

Line 39: Are all Myxococcota predatory or has it only been shown in the model organism *Myxococcus xanthus*?

Line 51: Which microbes do they prey on?

Line 78: I would remove the MAG that is only 38.32% as it's not very complete plus the fact that you've used a different model for completeness from all the other MAGs is questionable.

Line 88: It can be difficult to use AAI percentages as an indicator for whether the MAGs should be clustered at the class level as the range can be so varied. My suggestion would be to take representatives from different classes and genera and identify what the normal AAI range is for the genomes in the phylum Myxococcota. Otherwise you mention using GTDB-tk, you could use the RED values. Were the MAGs assigned a new class within GTDB-tk? And this is what is shown in Table S1.

Line 89: The correct etymology should be Kuafuibacteria, instead of Kuafubacteria. Also Kuafuibacterium and Houyibacterium, instead of Houyibacterium in Table S1. You should also provide etymology for the genus and species names as well. The genus names that you currently have in Table S1 doesn't make any sense except for the one ending with -bacterium; aestuarium is translated as tidal flat or estuarium, lacus is a lake, caenum is mud. It would be great if you can substitute those endings, for instance with -microbium, -bacter, -lacustris (belonging to a lake). This would give names such as Kuafuibacter, Kuafuimicrobium, Kuafuilacustris. You will need to designate (indicate) a type species of each new genus and type genome for each new species. You also need to propose Youwenlacus (correct to Youwenilacustris or Youwenibacter) and provide its etymology. It would be good to submit the names to the SeqCode.

Line 97: What is the percentage of predation in the Myxococcota?

Line 201: "...through an autonomously regulated phototrophic lifestyle."

Line 206 "lifestyle".

Line 220: How closely related are the genes?

Line 371: "...not only be a consumer shaping...may be a primary producer."

Line 376: "...single prokaryotic cells can serve as a player at multiple.."

Line 376: "rather than being confined to either decomposer, consumer or producer?"

Comments for methods

Line 408: Did you trim the adaptors from the sequencing reads?

Line 410: Remove the words "according to", you can just say "The sequence coverage of contigs was calculated by mapping the trimmed reads to each contig using BMap version 38.18.

Line 418: It says that the quantification of transcript expression was using salmon. But what did you map the metatranscriptomic sequencing reads too? Were they the myxococotal genomes or photosynthetic genes from the whole metagenome?

Line 452: Did you dRep your MAGs as well? Some of them are at 99-100% AAI (Fig S2).

Line 489: Do you mean queried?

Line 516: I'm not sure what is meant by rRNA genes. Did you find the rRNA genes and add them to the MSA?

Comments for figures and tables

Fig 1

Fig 1 is really difficult to see clearly. It would be better if you could zoom in on the Myxococcota.

Line 802: 1,713.

Fig 2

Line 824: They may also not have formed in clusters due to the fact that some of the MAGs have over 2,000 scaffolds.

Fig 3.

Which genomes are the representatives for each of the Myxococcota families? Are you using them all? What if the metabolic pathways are only found in half of the genomes? It would be good to have the number of genomes that you're using for the analysis in brackets next to each of the family names.

Fig 4

Were the expression data mapped back to the genomes? In 4A, you may need to point out that the green are photosynthesis genes and the grey are non-photosynthesis genes. Are there multiple Myxococcota genomes from the same study? Are they expressed evenly? In 4B, what is the representative genome of the family *Ca. Houyibacteriaceae*?

Line 860: Top highly expressed samples? Are the same levels of expression found in the other genomes?

Fig 5

Lifestyle

Fig S1

In the legend, the completeness is 32.4% to 100% and the contamination is 0% to 8.4%, however in Supplementary Table 1, the contamination is up to 9.33%. It also looks like there are genomes in the tree that have even more contamination. It's difficult to see the classes of interest in this figure. I strongly suggest creating trees of the classes of interest and only including the type species for the other genera plus two or three extra genomes. The genome with the long branch is a little worrying. I would remove that genome. Is it because it's very incomplete? Instead of using unphototroph, I would use non-phototroph. How did you include the 16S rRNA genes? It's very difficult to determine how large the circles are. I would remove the different sizes from the legend.

Fig S2

The AAI between many of the *Houyibacteriaceae* is 99-100%. I would consider these MAGs to be the same strain and would de-replicate them at least at the 95% ANI level. The same goes for some of the MAGs in the *Nannocystaceae*. It would be good to also include a phylogenetic tree next to the names so that we can also visually see how close they are. I now see in Table S1 that the MAGs have been dereplicated but you don't mention this in the main text. Are all the analyses performed on all the MAGs or the dereplicated ones?

Fig S3

It's difficult to know what the green squares were unless you zoom into the document at 300%. When I first looked at it I couldn't see green in the legend. I'm not sure if there's another way you can show the numbers that will be easier to see. Is dark green 1 copy and light green 2 copies?

Fig S4

You may want to check that your supplementary files are in order. Fig S4 is first mentioned in line 198 but Fig S5 and Fig S6 are mentioned on line 121.

Fig S5

I would remove the legend for the bootstraps as it's difficult to see how large the 0.9, 0.93, 0.95, 0.98 and 1 circles are.

Fig S6

1,201. Where are the Betaproteobacteria and why are they in brackets?

Fig S7

It took me a little while to understand what the different colours of the branches are. It maybe best to remove the brackets.

Fig S8

paper80

Table S2

Maybe to be consistent, just have the family name and the species name and remove the f__ for family. You should remove LLY-WYZ-19_1. Based on its completeness, this is probably why it's missing a lot of genes in the metabolism analysis.

Dear Reviewers,

Thank you for your careful reading and professional suggestions. All comments here have been taken into full consideration during the preparation of our revised manuscript. Below are our point-by-point responses to the comments. Page and line numbers are consistent with the revised manuscript. Responses of reviewers' questions are in blue color, quotations of new texts in the manuscript are in blown color.

Reviewer #1 (Remarks to the Author):

In this submission titled "Globally distributed phototrophic prokaryotic predators illuminate the origin and evolution of a novel chimeric lifestyle", Li and colleagues report their original finding that a diverse group of myxobacteria (recently classified as Myxococcota) encodes whole gene clusters known to enable photosynthetic activity. Given that several myxobacteria are known predators, they set out to demonstrate that the parallel presence of homologous genes tied to photosynthesis and heterotrophic predation mean that these species engage in chimeric life styles. As such, this finding would be highly intriguing and of high interest to a larger scientific community, especially for scientists in microbiology and botany. Yet, there are several major caveats that I will highlight further down below, which prevents this contribution to substantiate the claims it makes.

In more detail, the authors resorted to genome mining techniques to identify the presence of photosynthesis-related genes in metagenomes. They de novo assembled metagenomic reads (that are dubbed Metagenome-Assembled Genomes, or MAGs) and queried assemblies against a database containing an exhaustive list of relevant bacterial photosynthesis genes with parameters chosen to assure reasonable best matches (they also added a manual curation step of search results). Within the set of identified MAGs the authors quite surprisingly identified conserved photosynthesis-related gene clusters within the class of Myxococcota, including in several known families, but also in two entirely novel groups (to which they assign the novel names, Kuafubacteria, and Houyibacteriaceae). Given this observation, the authors would like to draw a strong connection between phototrophy and heterotrophic predation, the latter off which is a feature of many culturable myxobacteria, including *Myxococcus xanthus*.

The way they approach this, however, is shaky, and as such the connection between photosynthesis and predation remains highly speculative. Below I substantiate my concerns and give a verdict.

Response: We thank the reviewer for the constructive suggestion of our work. Based on the reviewer's comments, we removed predation analysis, reduced and toned down the description of connection between photosynthesis and predation throughout the revised manuscript. Instead, we focused more on photosynthesis analysis, and experimentally validated the functions of two important photosynthesis genes by heterologous expression.

The good: identification of photosynthetic-related gene clusters in myxobacteria

There is no question in my mind that what the paper found is completely new: the identification of photosynthetic-related gene clusters in MAGs, taxonomically classifiable as Myxococcota,

of which several representatives are model systems for multicellular development, sociality, and predation. I really like their phylogenetic approaches, these meet accepted standards, and their graphical representations are clear and aesthetically appealing. They also present phylogenetic evidence and gene expression data to further demonstrate that many or most of the complete sets of photosynthesis genes are conserved and expressed in nature. This approach seems very robust to me, and demonstrates that photosynthesis may be important in several myxobacterial clades.

Response: We thank the reviewer for the supportive comments.

The bad (I): the interpretation of active photosynthesis from expression data

No myxobacterial research group to date (including the great Dale Kaiser) has previously found evidence for photosynthesis in myxobacteria. As such, the presence of these gene clusters and even the knowledge of gene expression is surprising and very exciting. But in my mind, in itself it is not direct proof that myxobacteria actually perform photosynthesis (however likely this appears to be). It has been shown in molecular detail that *Myxococcus xanthus*, the most commonly used model system in the field, is able to respond to light (via photoreceptors) by producing pigments. This line of research has been conducted mainly by the group of Montserrat Elías-Arnanz (see for example: <https://doi.org/10.1126/science.aay1436>). I think it remains to be seen if the photosynthesis-related genes have maintained their original purpose in myxobacteria (over all the billions of years), or if they were repurposed for some other tasks. MAGs alone won't tell us.

Response: Thanks for the comment. Although the photosynthesis gene clusters (PGCs) from *Myxococcota* are complete and phylogenetically cluster together with the functional photosynthesis genes from *Proteobacteria*, we really agree with the reviewer that only analyzing MAGs cannot conclude whether these myxococcotal photosynthesis-related genes maintained their original purpose or were repurposed for other tasks. Thus, we performed heterologous expression of the chlorophyllide oxidoreductase genes *bchYZ* and phytoene desaturase gene *crtI* in mutants of the model purple bacterial phototroph *Rba. sphaeroides*, and we demonstrate that these *Myxococcota* photosynthesis genes do function as enzymes involved in bacteriochlorophyll and carotenoid biosynthesis. Nevertheless, we toned down the statements that are not experimentally tested by myxococcotal isolates. We here changed the phrase “phototrophic *Myxococcota*” to “potential phototrophic *Myxococcota* or *Myxococcota* with photosynthesis genes”, and added the description for limitations of this study on Line 402-405: “Current *in silico* analyses cannot directly prove that *Myxococcota* with PGC actually perform photosynthesis *in vivo*. Consequently, future pure cultivation of representatives of these *Myxococcota* will be required to validate the photosynthetic activity and hypotheses suggested in this study as well as other important capacities in myxococcotal biology.”

The bad (II): the conceptualisation of and lines of evidence for predation

The way the authors want to tie potential photosynthesis to potential predation is rather weak. This starts with the fact that it remains elusive what the authors think bacterial predation is, and what would be accepted hallmark genes involved in bacterial predation, especially for myxobacteria. Myxobacteria form multicellular groups in order to overwhelm their prey, but can

also kill by contact-dependance. While predatory life-styles are wide-spread across the myxobacteria, it is not yet clear if myxobacteria are generally predatory (see ref. 30 which the authors cited). In fact, many are saprophytes and also produce antimicrobials to fend off competitors, e.g., the cellulose-degrader *Sorangium cellulosum*, which the authors erroneously list as facultative predator.

While several killing mechanisms have been elucidated (eg, <https://doi.org/10.7554/eLife.72409>), just killing others is not predation. And, the general molecular basis of predation in myxobacteria is essentially still a mystery and the quest for elucidation of molecular elements has only recently begun.

To counter this knowledge gap, the authors resort to a machine-learning approach to infer common “predatory” genes. It is not entirely clear how this works to me, and so I am sceptical of the reported results., and this is mostly because of the list of genes it produces (see below).

These they report mainly in Tables S5 and S6, where they list what the authors define as “predators” (Table S5) and “predator-specific KEGG protein groups” (Table S6), respectively.

But not all members of Table S5 are predators, in fact *Sorangium cellulosum* is cellulolytic, as mentioned above, and can produces antimicrobial compounds meant for competitor control (killing) but not directly to use them as food source (they specialize on cellulose). Hence, any shared genes might not reflect predation per se, but simply antagonistic behaviors.

It is then not surprising, that Table S6 lists a lot of curious genes, which the authors call predation genes. Among them (on top) is listed *lacl*, the well characterized repressor of expression of genes encoding lactose degradation enzymes *lacZ*, *lacY* and *lacA*. *Lacl* is a gene that is also present and perhaps better understood in *E. coli*, not known to be predatory. The list also includes dozens of seemingly unspecific genes including ammonium transporters. To me it remains unclear what these genes represent, and how they can be labelled predation genes. And as such, I think this analysis is not conclusive.

Response: We thank the reviewer for the valuable comments. We agree with the comment that the result of machine learning prediction is not conclusive, especially under the background that the general molecular basis of predation in myxobacteria is still not clear, even though the “predation-related genes”¹⁻³ and the machine learning method for prediction of predatory behavior in *Myxococcota* have been reported by several groups^{4,5}. So, we decided to remove the analysis of “predation-prediction” in the revised manuscript as the reviewer suggested.

Our previous effort was to do a machine learning prediction at the whole genome level, and combined the “predator-specific KEGG protein groups” that was identified from another study³, to visualize the results. Therefore, it was not appropriate to indicate the names “predator-specific KEGG protein groups” because these KEGG protein groups also contain proteins that are not specific to predators. Moreover, based on the current background of myxobacterial biology, we cannot accurately list the “predator-specific KEGG protein groups” from the machine learning approach, since many of the protein groups are unspecific and related with

other functions or present in non-predatory microbes. We also thank the reviewer for pointing our error in labeling *Sorangium cellulosum* as predator (in fact they are cellulose-degrader).

In summary, we removed the analysis of “predation-prediction” in the revised manuscript. We are looking forward for future research that “predatory” genes and detailed mechanism are well resolved and more species are verified as predator or not. Thereafter, machine learning or other methods may be more indicative and convincing.

The ugly: ignoring well-established myxobacterial biology (which undermines their interpretation)

The most crucial issue arises from a lack of understanding of myxobacterial biology, which undermines the interpretation of the author's findings. Myxobacterial biology centers around two main pillars: multicellular development and social motility, both of which are clearly much better understood in molecular detail than predation is. Instead of using flagella to swim, most well-studied myxobacteria utilize two very different motility systems for movement across surfaces: social pili-driven motility (<https://doi.org/10.1038/s41467-020-18803-z>), and adventurous gliding motility (<https://doi.org/10.1073/pnas.1219982110>). Yet, in Figure 3 the authors depict the typical myxobacterium with a single flagellum not resembling anything that is usually studied in the myxobacterial community. What is more, the standard myxobacterial response to starvation is fruiting body formation in which the cells engage in highly coordinated multicellular development (and sticking to a precise developmental program). This part of the life cycle is extremely well studied, yet the authors only speculate that: “The phototrophic capability may be an important feature for the survival of predatory Myxococcota during intervals where prey is limited.” (LL. 185 – 186). In other words, they do not apply the more standard knowledge on development, but rather resort to a fictional myxobacterial model that fits their data best.

Response: Thank you very much for these valuable suggestions.

i) Myxobacteria have been paid much attention in the past decades because of their importance. Identifying their common features, such as multicellular development, adventurous and social motility, is still challenging only to use the data of MAGs, just as the reviewer's indication that “MAGs alone won't tell us”. Therefore, we deleted the single flagellum in the revised Figure 3 to avoid potential misleading after exploring the key genes for movement as described below.

First, to test the presence of flagella and pili related genes in the genomes of potential phototrophic *Myxococcota*, we used the web portal GhostKOALA on the KEGG website⁶. We found that the present pattern of pili related genes of potential phototrophic *Myxococcota* resembles that of cultured species. This inference is mainly based on the cluster pattern that many clades of potential phototrophic *Myxococcota* and cultured species clustered together (Response letter Fig. 1). For example, most of the genomes of potential phototrophic *Myxococcota* have PilY1 (essential for pilus extension in *Myxococcus xanthus*)⁷ and other genes that may be important for Type IV pili assembly. In addition, it seems likely that potential phototrophic *Myxococcota* contain more flagellar related genes compared with cultured species (Response letter Fig. 2). Except for the LLY-WYZ-1_1, LLY-WYZ-3_1, and LLY-WYZ-

15_3, other representatives of photosynthesis genes-containing *Myxococcota* clustered together, suggesting that gene groups of flagellar assembly of potential phototrophic *Myxococcota* are different compared with those of cultured myxobacterium. This is reasonable because many of the MAGs of potential phototrophic *Myxococcota* were collected from water-related environments, e.g., seawater, freshwater, and waste water, rather than soil environments. Under such environments, flagella might be very important for the movement of potential phototrophic *Myxococcota*.

In addition, we collected representative sequences of fruiting body formation (FruA, GenBank accession: ABF92619.1), social pili-driven motility (PilY1, GenBank accession: SDW47221.1), and adventurous gliding motility⁸ (AgIR, flagella stator homologs of MotA, GenBank accession: AAO22892.1; MotA, GenBank accession: QDF08298.1) from type species *Myxococcus xanthus*. We used these sequences as database to search similar proteins from the MAGs of potential phototrophic *Myxococcota* using blastp (coverage >75% and $e < 1 \times 10^{-20}$). We found that all MAGs of potential phototrophic *Myxococcota* contain AgIR and/or MotA, suggesting the potential of adventurous motility in these members (Response letter Fig. 3). Interestingly, the genes involved in fruiting body formation, adventurous and social motility were shared by the LLY-WYZ-3_1 and cultured *Myxococcus xanthus*. We further used these sequences to construct phylogenetic tree by employing IQ-Tree⁹. We found that the sequences of FruA, PilY1, and AgIR of LLY-WYZ-3_1 also shared close phylogenetic relationship with the sequences of *Myxococcus xanthus* (Response letter Fig. 5). These results suggest the metabolic potentials of fruiting body formation, adventurous and social motility of LLY-WYZ-3_1. However, the above annotation analyses were only based on the KEGG database or protein sequences from the cultured model organism *Myxococcus xanthus*⁶, hence, more experimental analyses need further comprehensive and systematic investigation to identify these important features in myxobacterial biology.

ii) To better interpret our findings, we added related knowledge in the introduction and refined our discussion with myxobacterial background knowledge throughout the revised manuscript. For example, in the introduction on Line 40-52, we added the description: "*Myxococcota* is an astonishing bacterial phylum because of their extraordinary social lifestyle (e.g., predation and fruiting body formation), which is unusual in prokaryotes." ... "Most species exhibit the abilities of adventurous motility and social motility. Predatory lifestyles have also been reported in many species of this phylum, and the "wolf pack" analogy has been widely used to describe their predatory behavior"..... "Some members of *Myxococcota* are saprophytic, and can produce antimicrobials to fend off competitors, e.g., the cellulose-degrading *Sorangium cellulosum*. Therefore, the capacity to produce a plethora of secondary metabolites for predation or competition makes *Myxococcota* a rich source for bioactive secondary metabolites for decades besides *Actinobacteria*, *Bacillus*, and fungi."

iii) We also added the description of fruiting body formation that is the standard myxobacterial response to starvation in the Discussion, e.g., on Line 216-218: "*Myxococcota* are known as social organisms with a predatory or saprophytic lifestyle, and many cultured strains can form fruiting bodies to resist extreme conditions, displaying flexible gene expression patterns that vary with environments."

Response letter Fig. 1 The presence and abundance of pili related genes in the representative genomes of potential phototrophic *Myxococcolta* and cultured myxobacteria.

Response letter Fig. 2 The presence and abundance of flagella related genes in the representative genomes of potential phototrophic *Myxococcota* and cultured myxobacteria.

Response letter Fig. 3 The heatmap depicts the blastp results using the predicted ORFs of potential phototrophic *Myxococcota* against the database of representative genes of fruiting body formation (FruA, GenBank accession: ABF92619.1), social pili-driven motility (PilY1, GenBank accession: SDW47221.1), and adventurous gliding motility (AgIR, GenBank accession: AAO22892.1; AgIR, homologs of MotA, GenBank accession: QDF08298.1) from

type species *Myxococcus xanthus*.

Response letter Fig. 5 The phylogenies of potential FruA, PiiY1, and AgIR (MotA) sequences of potential phototrophic *Myxococcota* and cultured myxobacterium.

The verdict:

In essence, the authors find something quite amazing, and I think then overreach by setting out to establish a link between the clear presence of photosynthesis-related genes in myxobacteria and the possibility of a chimeric life-style involving heterotrophic predation. In that respect they fail, so I think it is best for them to thoroughly reconsider the scope of their study, such that they can embed their incredibly interesting finding in a bullet-proof context. For that they need to stick to what can be reasonably shown using omics techniques alone, and incorporate well-established facts about myxobacterial biology that were gathered in the past around seventy years. The latter research has not found any evidence for photosynthesis in myxobacteria.

Response: Thank you for the inspiring verdict and constructive comments. We agree with the

reviewer's suggestions that only using a model of machine learning-based "predation prediction" without *in vivo* verification to link highly speculative predation to our findings of photosynthesis-related genes in myxobacteria is not sufficiently convincing. We also note that the accuracy and magnitude of training set as well as the performance of the resulting model of machine learning-based "predation prediction" in the previous version of our manuscript are limited. Thus, we have removed the result of "predation prediction", added the heterologous expression validating the functions of photosynthesis genes, and re-discussed the findings in this paper closely centered on omics data and followed the current myxobacterial biological context to the best of our knowledge.

Specific comments:

- The need to tone down any bold statement that remain unsubstantiated, (e.g., LL. 113-114: for the phototrophic Myxococcota)

Response: Thanks for the valuable suggestion. We toned down the statements that remain unsubstantiated, e.g., "phototrophic *Myxococcota*" to "potential phototrophic *Myxococcota*" or "MAGs of *Myxococcota* with photosynthesis genes".

- The very book chapter the authors cite as source that claims that all myxobacteria are predatory (ref. 30) states the following "It is not yet known whether all myxobacteria are predatory, however it seems likely that it is a common feature of the order; predators have been isolated from all three myxobacterial sub-orders, including representatives from the Chondromyces, Corallococcus, Enhygromyxa, Myxococcus, Plesiocystis, Pyxidicoccus, Racemicystis, Sorangium and Stigmatella genera (Amiri Moghaddam et al. 2016, 2018; Awal et al. 2016; Livingstone et al. 2017; Perez et al. 2016)." So, general predation is not verified truth but rather needs to be properly tested.

Response: Thanks for the suggestion. We deleted the analysis of machine learning based prediction and toned-down the relevant description of "general predation" in the revised manuscript.

- LL. 72-73 missing information what is meant by «near» completeness of (bacterio)chlorophyll biosynthesis pathways.

Response: Thanks for the suggestion. We modified the description in the Result on Line 81-82: "A total of 8,221 genomes containing a partial to complete (gene prevalence > 50%) (bacterio)chlorophyll biosynthesis pathway were retained." We also added parameter to the Methods on Line 452-453: "Genomes containing a partial to complete (bacterio)chlorophyll biosynthesis pathway were screened as potential chlorophototrophs."

- L. 131 "The highly similar topologies of photosynthesis gene trees". To make that statement, did the authors perform a statistical test to test for (in)congruence of the tree topologies (eg, using a Kishino–Hasegawa test)?

Response: Thanks for the suggestion. We performed tree topology comparison to check the congruence using *Myxococcota* MAGs with complete photosynthesis gene cluster (Response letter Table 1). The trees were all constructed by using IQ-tree with best-fitting models plus C60 model. The result of both Kishino-Hasegawa test¹⁰ and Shimodaira-Hasegawa test¹¹ using the key photosynthesis marker type II reaction center PufLM and species trees using two different sets of marker genes showed that their topologies are not significantly different

(p -value > 0.05). We also visualized these trees in the Response letter Fig. 5 to intuitively compare tree topologies. We found that the tree topologies are consistent (both at the class level and at the family level).

Response letter Table 1. Tree topologies test for the congruence of photosynthesis gene trees and species trees.

Type II photosynthesis reaction center	Different tested marker sets	logL	p-value Kishino-Hasegawa test	p-value Shimodaira-Hasegawa test
PufLM	37 markers	-4840.471955	0.0508	0.0935
	GTDB markers	-4831.906993	0.0507	0.0935
	BchHDI	-4831.907181	0.154	0.342
	BchLNB	-4839.372352	0.0224	0.104
	PGC	-4823.738882	0.154	0.342

Response letter Fig. 5 Phylogenies of photosynthesis gene trees and species trees using representative genomes.

- Figure 1:
 - o add color information to those taxa that are non-phototrophic (I think the color used is a shade of brown).
 - o better highlight all taxa that belong to the myxobacteria in the trees (ie, by a clearer line of

demarkation and circle sector enclosing all *Myxococcota*.

Response: Thanks for the comments. We re-visualized the Figure 1 as the reviewer suggested. In the inner ring of the Figure 1, we used a shade of brown to indicate the lineages belonging to *Myxococcota*. We also used a green background to indicate phototrophic lineages, and used grey background to indicate non-phototrophic lineages. In addition, we added the description of caption on Line 925-929: "In the inner ring, lineages of *Myxococcota* are shaded with brown background. In the middle ring, phototrophic lineages are colored with green background, and non-phototrophic lineages are colored with grey background. The outer ring of phylogeny is colored based on different types of reaction centers."

Revised Figure 1. Phylogenetic affiliations of the potential phototrophic *Myxococcota* MAGs, and corresponding PufLM and BchHDI protein sequences. (A) Phylogenomic affiliation of the MAGs based on 37 conserved protein sequences and using 232 representative reference bacterial genomes. Alignments were based on MAFFT and then filtered with trimAl. The tree was built using IQ-Tree with 1,000 bootstrap replicates. The bootstrap supporting values above 0.9 were indicated with solid circles. The phylogenetic tree was rooted at the midpoint. In the inner ring, lineages of *Myxococcota* are shaded with brown background. In the middle ring, phototrophic lineages are colored with green background, and non-phototrophic lineages are

colored with grey background. The outer ring of phylogeny is colored based on different types of reaction centers. All phyla are labeled near the phylogenies, with the exception of *Proteobacteria* that are labeled based on different classes, e.g., *Alphaproteobacteria* and *Gammaproteobacteria* (*Betaproteobacteria* shared same color with *Gammaproteobacteria* based on the taxonomy in GTDB r207). Six potential phototrophic families of *Myxococcota* are also labeled with different colors near their phylogenies. (B) The phylogenetic trees were constructed based on the alignments of PufLM with 571 aligned positions and BchHDI with 1,702 aligned positions. Alignments were based on MAFFT and then filtered with trimAl, and the trees were built using IQ-Tree with 1,000 bootstrap replicates. The bootstrap supporting values above 0.9 are indicated with solid circles. Lineages of *Myxococcota* are colored with brown for background. Members from *Alphaproteobacteria*, *Gammaproteobacteria* (including *Betaproteobacteria*), *Gemmatimonadota*, and six families of *Myxococcota* are assigned different background colors. The outer ring of phylogeny is colored based on corresponding phylum.

Reviewer #2 (Remarks to the Author):

The study by Li et al reports the first discovery of chlorophototrophic members of the predatory bacterial phylum Myxococcota and expands the number of phototrophic bacterial phyla to nine. They show the photosynthetic genes are from a monophyletic origin and also show vertical evolution. The discovery represents the first time a chimeric lifestyle of photosynthesis and predation are integrated into a single prokaryotic cell.

They assembled genomes from the NCBI prokaryotic database and the Earth's Microbiome catalog, which was used as queries for screening photosynthesis related homologs resulting in the discovery of the Myxococcota MAGs. They created gene trees from the photosynthetic genes to compare them with genes from other phyla and inferred metabolic schemes based on the presence of genes. They also use metatranscriptional data to determine the expression of photosynthesis genes for the different samples and propose the potential chimeric lifestyle of Myxococcota which involves photosynthesis, heterotrophy and predation.

The manuscript reads well and I think the work will be a significant addition to the field, especially in regards to the evolution of photosynthesis. The work supports the conclusions and claims. Below I have listed comments to address. The methods are sound and the work meets the expected standards in the field.

Response: We thank the reviewer for the supportive comments.

Comments for the main text

Line 39: Are all Myxococcota predatory or has it only been shown in the model organism *Myxococcus xanthus*?

Response: Predation has been verified in many genera (Response letter Table 2), e.g., *Coralloccoccus*, *Myxococcus*, and *Stigmatella*. Although predatory capacity seems likely to be a common feature in cultured strains of *Myxococcota*, we cannot conclude that myxobacteria are generally predatory, just as the comment from Reviewer #1. Meanwhile, given that the general molecular basis of predation in myxobacteria remains unclear, the result of predation-prediction based on machine learning (most phototrophic members are also predicted as predators) is not cogent enough. Therefore, even if the "predation-related genes"^{1-3,12} and the machine learning method for prediction of predation behavior in *Myxococcota* have been reported by several groups^{4,5}, we deleted the machine learning prediction and toned down the discussion of "the lifestyle depends on photosynthesis and predation of *Myxococcota*" as suggested by Reviewer #1.

Response letter Table 2 The information of selected predatory *Myxococcota* genomes.

NCBI Taxonomy of cultured genomes				
Accession_Number	NCBI_Taxonomy	StrainName	PredationType	References
GCA_00012685.1	Myxococcus xanthus	DK 1622	Predator	Goldman, B. S. et al. Evolution of sensory complexity recorded in a myxobacterial genome. Proc Natl Acad Sci U S A 103, 15200–15205, doi:10.1073/pnas.0607335103 (2006).
GCA_000224805.1	Halangium ochraceum	DSM 14385	Predator	Fudou, R., Jojma, Y., Izuka, I., & Amanaka, S. Halangium ochraceum gen. nov., sp. nov. and Halangium tepidum sp. nov.: novel moderately halophilic myxobacteria isolated from a coastal area. Microbiol. Dr. 157, 1–10, doi:10.1093/mic/duu187 (2015).
GCA_000165485.1	Stigmatella aurantia	DW42-1	Predator	Reichenbach, H. & Dworkin, M. Studies on Stigmatella aurantia (Myxobacterales). Journal of General Microbiology 58, 3–14, doi:10.1099/00222720-58-1-3 (1969).
GCA_000170095.1	Plesiocystis pacifica	SR-1	Predator	Izuka, T. et al. Plesiocystis pacifica gen. nov., sp. nov., a marine myxobacterium that contains dihydrogenated menaquinone, isolated from the Pacific coasts of Japan. Int J Syst Evol Microbiol 56, 109–115, doi:10.1093/ismj/56.1.109 (2006).
GCA_000252295.1	Coralococcus coralifides	DSM 2259	Predator	Livingstone, P. G., Morphey, R. M. & Whitworth, D. E. Genome Sequencing and Pan-Genome Analysis of 23 Coralococcus spp. Strains Reveal Unexpected Diversity, With Par
GCA_000737315.1	Halangium minutum	DSM 14724	Predator	Sharma, G., Khatri, I. & Subramanian, S. Comparative Genomics of Myxobacterial Chemotaxis Systems. J Bacteriol 200, doi:10.1128/JB.00620-17 (2018).
GCA_000737325.2	Sandaracium amylolyticus	DSM 53668	Predator	Mohr, K. I., Garcia, R. O., Gerth, K., Irshchik, H. & Muller, R. Sandaracium amylolyticus gen. nov., sp. nov., a starch-degrading soil myxobacterium, and description of Sandaracium
GCA_001931535.1	Mincycystis rosea	DSM 24000	Predator	Garcia, R., Gempeler, K. & Muller, R. Mincycystis rosea gen. nov., sp. nov., a polyunsaturated fatty acid-rich and steroid-producing soil myxobacterium. Int J Syst Evol Microbiol 56, 109–115, doi:10.1093/ismj/56.1.109 (2006).
GCA_002343915.1	Nannocystis exedens	DSM 71	Predator	Reichenbach, H. Nannocystis exedens gen. nov., spec. nov., a new myxobacterium of the family Sorangiaceae. Arch Mikrobiol 70, 119–138, doi:10.1007/BF00412203 (1970).
GCA_002994615.1	Enhygromyxa salina	SWB005	Predator	Izuka, T. et al. Enhygromyxa salina gen. nov., sp. nov., a slightly halophilic myxobacterium isolated from the coastal areas of Japan. Syst Appl Microbiol 26, 189–196, doi:10.1016/S1151-7755(03)00012-5 (2003).
GCA_000109545.1	Stigmatella aurantia	DSM 17044	Predator	Reichenbach, H. & Dworkin, M. Studies on Stigmatella aurantia (Myxobacterales). Journal of General Microbiology 58, 3–14, doi:10.1099/00222720-58-1-3 (1969).
GCA_003260125.1	Lujinxingia litoralis	B210	Predator	Guo, L. Y., Li, C. M., Wang, S., Mu, D. S. & Du, Z. J. Lujinxingia litoralis gen. nov., sp. nov. and Lujinxingia sediminis sp. nov., two new representatives in the order Bradymonadales
GCA_003367535.1	Bradymonadaceae bacterium	TMQ3	Predator	Mu, D. S. et al. Bradymonadaceae, a novel bacterial predator group with versatile survival strategies in saline environments. Microbiome 8, 126, doi:10.1186/s40168-020-00900-0 (2020).
GCA_004005565.1	Lujinxingia sediminis	SEH01	Predator	Guo, L. Y., Li, C. M., Wang, S., Mu, D. S. & Du, Z. J. Lujinxingia litoralis gen. nov., sp. nov. and Lujinxingia sediminis sp. nov., two new representatives in the order Bradymonadales
GCA_004362595.1	Bradymonas sediminis	FA350	Predator	Wang, Z. J., Liu, Q. Q., Zhao, L. H., Du, Z. J. & Chen, G. J. Bradymonas sediminis gen. nov., sp. nov., isolated from coastal sediment, and description of Bradymonadaceae fam
GCA_008517175.1	Persicimonas caeni	YN101	Predator	Wang, S., Mu, D. & Du, Z. J. Persicimonas caeni gen. nov., sp. nov., the Representative of a Novel Wide-Ranging Predatory Taxon in Bradymonadales. Frontiers in microbiology
GCA_007993755.1	Microvenator marinus	V1718	Predator	Wang, S., Chen, G. J. & Du, Z. J. Microvenator marinus gen. nov., sp. nov., isolated from marine sediment, and description of Microvenatoraceae fam. nov. and Lujinxingiaceae
GCA_007994975.1	Bradymonadales bacterium	TMG1	Predator	Mu, D. S. et al. Bradymonadaceae, a novel bacterial predator group with versatile survival strategies in saline environments. Microbiome 8, 126, doi:10.1186/s40168-020-00900-0 (2020).
GCA_007997005.1	Lujinxingia vulgaris	TMQ2	Predator	Wang, S., Mu, D.-S., Li, G.-Y. & Du, Z.-J. Description of Lujinxingia vulgaris sp. nov., isolated from coastal sediment via prey-traps. Antonie van Leeuwenhoek 114, 1805–1818, doi:10.1007/s12220-019-00120-0 (2019).
GCA_007997015.1	Lujinxingia vulgaris	TMQ4	Predator	Wang, S., Mu, D.-S., Li, G.-Y. & Du, Z.-J. Description of Lujinxingia vulgaris sp. nov., isolated from coastal sediment via prey-traps. Antonie van Leeuwenhoek 114, 1805–1818, doi:10.1007/s12220-019-00120-0 (2019).
GCA_000087165.1	Sorangium cellulosum B	So ce 56	Cellulose-degrader	Gerth, K., Bedorf, N., Hofle, G., Irshchik, H. & Reichenbach, H. Epithilons A and B: antifungal and cytotoxic compounds from Sorangium cellulosum (Myxobacterales). Production, f
GCA_000013385.1	Anaeromyxobacter dehalogenans	ZCP-C	Non predator	Thomas, S. H. et al. The mosaic genome of Anaeromyxobacter dehalogenans strain ZCP-C suggests an aerobic common ancestor to the delta-protobacteria. PLoS One 3, e2711 (2008).
GCA_001263175.1	Vulgabacter incomplus	DSM 27710	Non predator	Yamamoto, E., Muramatsu, H. & Nagai, K. Vulgabacter incomplus gen. nov., sp. nov. and Labitrix luteola gen. nov., sp. nov., two myxobacteria isolated from soil in Yakushiji
GCA_001263205.1	Labitrix luteola	DSM 27648	Non predator	Yamamoto, E., Muramatsu, H. & Nagai, K. Vulgabacter incomplus gen. nov., sp. nov. and Labitrix luteola gen. nov., sp. nov., two myxobacteria isolated from soil in Yakushiji
GCA_001931505.1	Pajarofellbacter abortibovis	BTF92-0548A/99-0131	Non predator	Welby, B. T. et al. Genome Report: Identification and Validation of Antigenic Proteins from Pajarofellbacter abortibovis Using De Novo Genome Sequence Assembly and Reverse

Line 51: Which microbes do they prey on?

Response: Predatory species of *Myxococcota* are able to feed on a broad range of microbes like soil bacteria¹³, pathogens^{14,15}, Cyanobacteria¹⁶, and fungi, including yeast^{15,17,18}.

Line 78: I would remove the MAG that is only 38.32% as it's not very complete plus the fact that you've used a different model for completeness from all the other MAGs is questionable.

Response: Thanks for the suggestion. We have removed this MAG.

Line 88: It can be difficult to use AAI percentages as an indicator for whether the MAGs should be clustered at the class level as the range can be so varied. My suggestion would be to take representatives from different classes and genera and identify what the normal AAI range is for the genomes in the phylum Myxococcota. Otherwise you mention using GTDB-tk, you could use the RED values. Were the MAGs assigned a new class within GTDB-tk? And this is what is shown in Table S1.

Response: Thanks for your valuable suggestions. i) The MAGs that we named *Candidatus* (*Ca.*) *Kuafubacteria* are assigned to the class with placeholder name c_WYAZ01 using GTDB-Tk. This class has six representative genomes in the GTDB r207 (the condition is consistent in the recently updated GTDB r214) (Response letter Table 3). ii) We added representative genomes from different classes and genera in *Myxococcota* to the AAI and ANI analyses to indicate the taxonomic level in the revised Supplementary Figure S1. The results showed that *Ca.* *Kuafubacteria* shared ~45–49 AAIs and ~61–72 ANIs with other classes. Such a result is similar compared with other classes. Indeed, AAI and ANI values are so varied. iii) Thus, we also placed the phylogenomic tree near the heatmap to indicate the taxonomic level according to your suggestion. Comprehensively, by combing the results of phylogenetic trees, GTDB-Tk classification, AAI and ANI analyses, we determined that these MAGs should be clustered at the class level. iv) We also checked the RED values using the software phylorank (<https://github.com/dparks1134/PhyloRank>). The RED analyses do not provide additional supports because the six genomes of *Ca.* *Kuafubacteria* belong to a same family and currently other genomes from different families (and orders) of *Ca.* *Kuafubacteria* are unavailable. v) To facilitate comparisons for placeholder names and the names we proposed, we added the taxonomy assignment of GTDB-Tk in the revised Table S1.

Response letter Table 3 NCBI and GTDB r207 taxonomy of the novel MAGs from *Ca. Kuafubacteria* (c_WYAZ01). The screenshot was generated from the GTDB website (<https://gtdb.ecogenomic.org/>).

Accession	NCBI organism name	NCBI taxonomy	GTDB taxonomy	GTDB species representative	NCBI type material
GCA_011526105.1	Deltaproteobacteria bacterium	d__Bacteria; p__Proteobacteria; c__Deltaproteobacteria; o__f__g__s__	d__Bacteria; p__Myxococcota; c__WYAZ01; o__WYAZ01; f__WYAZ01; g__WYAZ01; s__WYAZ01 sp011526105	yes	
GCA_016703535.1	Deltaproteobacteria bacterium	d__Bacteria; p__Proteobacteria; c__Deltaproteobacteria; o__f__g__s__	d__Bacteria; p__Myxococcota; c__WYAZ01; o__WYAZ01; f__WYAZ01; g__JADJBV01; s__JADJBV01 sp016703535	yes	
GCA_016713425.1	Deltaproteobacteria bacterium	d__Bacteria; p__Proteobacteria; c__Deltaproteobacteria; o__f__g__s__	d__Bacteria; p__Myxococcota; c__WYAZ01; o__WYAZ01; f__WYAZ01; g__JADJPE01; s__JADJPE01 sp016713425	yes	
GCA_019634385.1	Myxococcales bacterium	d__Bacteria; p__Proteobacteria; c__Deltaproteobacteria; o__Myxococcales; f__g__s__	d__Bacteria; p__Myxococcota; c__WYAZ01; o__WYAZ01; f__WYAZ01; g__JAHQVK01; s__JAHQVK01 sp019634385	yes	
GCA_020430745.1	Myxococcales bacterium	d__Bacteria; p__Proteobacteria; c__Deltaproteobacteria; o__Myxococcales; f__g__s__	d__Bacteria; p__Myxococcota; c__WYAZ01; o__WYAZ01; f__WYAZ01; g__JACKEN01; s__JACKEN01 sp020632995		
GCA_020632995.1	Deltaproteobacteria bacterium	d__Bacteria; p__Proteobacteria; c__Deltaproteobacteria; o__f__g__s__	d__Bacteria; p__Myxococcota; c__WYAZ01; o__WYAZ01; f__WYAZ01; g__JACKEN01; s__JACKEN01 sp020632995	yes	

Rows per page: 10 1-6 of 6 < >

Line 89: The correct etymology should be *Kuafuibacteria*, instead of *Kuafubacteria*. Also *Kuafuibacterium* and *Houyibacterium*, instead of *Houyibacterium* in Table S1. You should also provide etymology for the genus and species names as well. The genus names that you currently have in Table S1 doesn't make any sense except for the one ending with -bacterium; *aestuarium* is translated as tidal flat or estuarium, *lacus* is a lake, *caenum* is mud. It would be great if you can substitute those endings, for instance with -microbium, -bacter, -lacustris (belonging to a lake). This would give names such as *Kuafuibacter*, *Kuafuimicrobium*, *Kuafuilacustris*. You will need to designate (indicate) a type species of each new genus and type genome for each new species. You also need to propose *Youwenlacus* (correct to *Youwenilacustris* or *Youwenibacter*) and provide its etymology. It would be good to submit the names to the SeqCode.

Response: Thanks very much for this valuable comment. i) We have checked the etymology and changed all of the misassigned endings, e.g., *Kuafucaenum* to *Kuafucaenimonas*, *Deshenbium* to *Xihebacterium*. All changed names can be found in the Supplementary Table S1. ii) In addition, we found that in the case of "mythical people", it seems likely that it is not recommended to add '-i-' to the end of the prefixes (Response letter Fig. 6). iii) We also added the description of type species of each new genus and type genome for each new species and related etymology on Line 70-360 in the Supplementary Notes, according to your suggestion. iv) It is very important to submit the names to the SeqCode, and we have uploaded the information to SeqCode. Nevertheless, the name with published work and genome accessions linked to NCBI are required to complete the process of submitting names to SeqCode, so we will finish the complete process once the genomes were published.

Redacted

Line 97: What is the percentage of predation in the Myxococcota?

Response: Predatory capacity seems likely to be a common feature in cultured strains of *Myxococcota* (Table S1), but we cannot conclude that myxobacteria are generally predatory. In our previous version of the manuscript, we applied a machine learning method to predict the predatory behavior, and the results showed that most of the potential phototrophic *Myxococcota* may be predators simultaneously (32/33). The training set of the machine learning method was based on the genomes of predatory (20) or non-predatory species (5) that have been experimentally verified according to published papers (Table S4). The number of predatory species may have been underestimated because many earlier studies did not submit sequences of predatory genomes to NCBI. Thus, the percentage of predatory behavior in cultured species of *Myxococcota* might exceed 80% (20/25). Whereas several other studies inferred that some uncultured species of *Myxococcota* are non-predatory or non-social, for example, potential non-predatory species of the class c__UBA4248⁵ and potential non-social species of the order o__JAFGXQ01⁴. Nevertheless, as suggested by Reviewer #1, the predatory genes are still not clear, and detailed predatory mechanism is also unknown, therefore, we deleted relevant analyses and toned down the description of this intriguing metabolic type.

Line 201: "...through an autonomously regulated phototrophic lifestyle."

Response: Thanks for this suggestion. We changed the "autonomously" to "flexibly" on Line 235-236: "...allow them to cope with survival challenges through flexibly regulated phototrophic lifestyle".

Line 206 "lifestyle".

Response: Checked.

Line 220: How closely related are the genes?

Response: The protein sequences of the genes showed ~60–80% identities to the sequences of *Proteobacteria* and *Gemmatimonadota* (Response letter Table 4).

Response letter Table 4 The blast results of myxococcotal BchE-like proteins from the NCBI website (<https://blast.ncbi.nlm.nih.gov/Blast.cgi>).

Description	Max Score	Total Score	Query Cover	E value	Per. Ident	Acc. Len	Accession
bchE-Myxococcaceae-LLY-WYZ-3_1							
magnesium-protoporphyrin IX monomethyl ester anaerobic oxidative cyclase [Gemmatimonadota bacterium]	806	806	99%	0	76.66%	560	MCA9737111.1
magnesium-protoporphyrin IX monomethyl ester anaerobic oxidative cyclase [Roseiflexus sp.]	742	742	99%	0	69.31%	527	MCS6841723.1
magnesium-protoporphyrin IX monomethyl ester anaerobic oxidative cyclase [Roseiflexus castenholzii]	741	741	99%	0	69.11%	527	WVP_012120105.1
magnesium-protoporphyrin IX monomethyl ester cyclase [Roseiflexus sp.]	741	741	99%	0	69.11%	527	GIW00575.1
magnesium-protoporphyrin IX monomethyl ester anaerobic oxidative cyclase [Roseiflexus sp.]	741	741	99%	0	68.90%	528	MB09336467.1
magnesium-protoporphyrin IX monomethyl ester anaerobic oxidative cyclase [Roseiflexus castenholzii]	740	740	99%	0	68.90%	527	PMP76421.1
bchE-Kuafubacteriaceae-LLY-WYZ-17_1							
magnesium-protoporphyrin IX monomethyl ester anaerobic oxidative cyclase [Gemmatimonadota bacterium]	825	825	98%	0	78.30%	560	MCA9737111.1
magnesium-protoporphyrin IX monomethyl ester anaerobic oxidative cyclase [Gemmatimonadota bacterium]	746	746	98%	0	71.20%	549	MCA0375597.1
magnesium-protoporphyrin IX monomethyl ester anaerobic oxidative cyclase [Gemnicoccaceae bacterium]	745	745	99%	0	68.95%	564	TVQ35446.1
magnesium-protoporphyrin IX monomethyl ester anaerobic oxidative cyclase [Burkholderiales bacterium]	743	743	98%	0	69.23%	581	MCA3213571.1
magnesium-protoporphyrin IX monomethyl ester anaerobic oxidative cyclase [Burkholderiales bacterium]	742	742	98%	0	69.03%	581	MCA3222762.1
magnesium-protoporphyrin IX monomethyl ester anaerobic oxidative cyclase [Burkholderiales bacterium]	741	741	98%	0	69.15%	599	MCA3219156.1
bchE-Polyangiaceae-LLY-WYZ-9_1							
magnesium-protoporphyrin IX monomethyl ester anaerobic oxidative cyclase [Gemmatimonadota bacterium]	822	822	99%	0	77.73%	560	MCA9737111.1
magnesium-protoporphyrin IX monomethyl ester anaerobic oxidative cyclase [Burkholderiales bacterium]	760	760	99%	0	69.82%	581	MCA3216024.1
magnesium-protoporphyrin IX monomethyl ester anaerobic oxidative cyclase [Burkholderiales bacterium]	759	759	99%	0	69.82%	581	MCA3213571.1
magnesium-protoporphyrin IX monomethyl ester anaerobic oxidative cyclase [Burkholderiales bacterium]	759	759	99%	0	69.82%	581	MCA3222762.1
magnesium-protoporphyrin IX monomethyl ester anaerobic oxidative cyclase [Burkholderiales bacterium]	759	759	99%	0	70.02%	581	MCA3229655.1
magnesium-protoporphyrin IX monomethyl ester anaerobic oxidative cyclase [Gemnicoccaceae bacterium]	758	758	100%	0	70.34%	564	TVQ35446.1
bchE-Houyibacteriaceae-LLY-WYZ-14_1							
magnesium-protoporphyrin IX monomethyl ester anaerobic oxidative cyclase [Gemmatimonadota bacterium]	771	771	98%	0	72.82%	560	MCA9737111.1
magnesium-protoporphyrin IX monomethyl ester anaerobic oxidative cyclase [Burkholderiales bacterium]	729	729	98%	0	68.15%	599	MCA3219156.1
magnesium-protoporphyrin IX monomethyl ester anaerobic oxidative cyclase [Hypomicrobium sp.]	729	729	98%	0	68.50%	532	NOT69919.1
magnesium-protoporphyrin IX monomethyl ester anaerobic oxidative cyclase [Burkholderiales bacterium]	726	726	98%	0	67.54%	581	MCA3216024.1
magnesium-protoporphyrin IX monomethyl ester anaerobic oxidative cyclase [Burkholderiales bacterium]	724	724	98%	0	67.34%	581	MCA3224907.1
magnesium-protoporphyrin IX monomethyl ester anaerobic oxidative cyclase [Burkholderiales bacterium]	723	723	98%	0	67.34%	581	MCA3213571.1

Line 371: “..not only be a consumer shaping...may be a primary producer.”

Response: Thanks for this suggestion. The sentence was changed on Line 383-384: “*Myxococcota* are not only consumers shaping microbial communities by means of top-down control, but also may be primary producers.”

Line 376: “..single prokaryotic cells can serve as a player at multiple..”

Response: This sentence was removed.

Line 376: “rather than being confined to either decomposer, consumer or producer?”

Response: The sentence was deleted.

Comments for methods

Line 408: Did you trim the adaptors from the sequencing reads?

Response: Thanks for this comment. The adaptors were trimmed with the software Scythe (<https://github.com/vsbuffalo/scythe>).

Line 410: Remove the words “according to”, you can just say “The sequence coverage of contigs was calculated by mapping the trimmed reads to each contig using BMap version 38.18.”

Response: Thanks for this suggestion. We changed the sentence on Line 436-437: “The sequence coverage of contigs was calculated by mapping the trimmed reads to each contig using BMap version 38.18.”

Line 418: It says that the quantification of transcript expression was using salmon. But what did you map the metatranscriptomic sequencing reads too? Were they the myxococcotal genomes or photosynthetic genes from the whole metagenome?

Response: Here we used the salmon software for quantification of transcript expression (<https://doi.org/10.1038/nmeth.4197>). We merged six representative myxococcotal genomes into a single file, and mapped the metatranscriptomic sequencing reads to this file. The results

are shown in the Fig. 5. In addition, to indicate the PGC expression pattern in a genome, we also visualized the expression result of the genome LLY-WYZ-15_3 of *Ca. Houyibacteriaceae* so that we can identify the potential expression of an entire PGC from a single genome. This result is shown in the Figure 5b. Relevant description was added in the Methods on Line 520-527: “For the quantification of transcript expression, we used salmon version 1.9.0 to map the trimmed metatranscriptomic reads to the six representative *Myxococcota* genomes. The abundance of each gene in metatranscriptomes was calculated as the metric-FPKB (fragments per kilobase per billion metatranscriptomic fragments) normalized based on the gene length and sequencing depth. For the profile of global expression of *Myxococcota* photosynthesis genes, we summed the FPKB of genes that belongs to the same function to calculate the functional abundance of each metatranscriptomic library.”

Line 452: Did you dRep your MAGs as well? Some of them are at 99-100% AAI (Fig S2).

Response: Thanks for this suggestion. We did dRep analysis for the 32 MAGs. The dRep result was used for the analyses of PGC (Fig. 2c, Fig. S4, S8), metabolisms (Fig. 3), gene expressions (Fig. 4 and Fig.5), AAI and ANI (revised Fig. S1). In order to provide more detailed information, the sequences of all MAGs were used in the phylogenetic trees (Fig. 1, Fig. S2, S3, S5, S6, S9, S10) and genome comparisons (Fig. 2ab and Fig. S7).

Line 489: Do you mean queried?

Response: Corrected. See Line 488.

Line 516: I'm not sure what is meant by rRNA genes. Did you find the rRNA genes and add them to the MSA?

Response: Thanks for this suggestion. We have deleted the unmatched description of “rRNA genes”, as we originally intended to say the ribosome RNA protein sequences. The revised sentence is on Line 513-515: “...the 32 MAGs and all high-quality *Myxococcota* genomes available from the GTDB r207 were used to construct phylogenetic trees based on the concatenated alignments of 120 conserved single-copy marker genes using GTDB-Tk.”

Comments for figures and tables

Fig 1

Fig 1 is really difficult to see clearly. It would be better if you could zoom in on the *Myxococcota*.

Response: Thanks for this comment. We re-visualized the figure by simplifying dozens of colors for different phyla. We hope this new version is clear now. In the inner ring, we used a shade of brown to indicate the lineages belonging to *Myxococcota*. We also used a green background to indicate phototrophic lineages, and used grey background to indicate non-phototrophic lineages. We also added the description of caption on Line 925-929: “In the inner ring, lineages of *Myxococcota* are shaded with brown background. In the middle ring, phototrophic lineages are colored with green background, and non-phototrophic lineages are colored with grey background. The outer ring of phylogeny is colored based on different types of reaction centers.”

Line 802: 1,713.

Response: Revised.

Fig 2

Line 824: They may also not have formed in clusters due to the fact that some of the MAGs have over 2,000 scaffolds.

Response: Thanks for this comment. We added the description on Line 953-956: "Photosynthesis genes did not form apparent PGC (photosynthesis gene number <12 in any clusters) for the remaining 11 MAGs (average completeness ~84.6%), potentially attributed to incomplete assembly **and/or binning** of fragment reads in the next generation sequencing."

Fig 3.

Which genomes are the representatives for each of the Myxococcota families? Are you using them all? What if the metabolic pathways are only found in half of the genomes? It would be good to have the number of genomes that you're using for the analysis in brackets next to each of the family names.

Response: Thanks for this comment.

i) Representatives of six *Myxococcota* families used for metabolic visualization are LLY-WYZ-15_3, LLY-WYZ-17_1, LLY-WYZ-3_1, LLY-WYZ-12_1, LLY-WYZ-9_1, and LLY-WYZ-13_1. The accessions, taxonomy, and related information of representative genomes were also listed in the revised Supplementary Table S1. The selection of representative myxococcotal genomes for metabolic visualization was mainly based on their genome quality and completeness of metabolic pathways (see Line 572-585).

ii) We note that all of these genomes have complete bacteriochlorophyll synthesis pathways, but the completeness of other metabolic pathways varies with genomes. So, we used the half-filled parts in snowflake to indicate the absence of genes in the representative genome but present in other genomes of the same family. The genomes that have additional genes to supplement the representatives are labeled as "Additional" genome. For example, representative LLY-WYZ-9_1 of the Polyangiaceae contains complete assimilatory sulfate reduction (ASR) pathway but lack the genes of dissimilatory nitrate reduction (DNR), while the LLY-WYZ-7_1 from the same family contains complete DNR pathway. In this case the DNR pathway would be filled with half parts in snowflake to show the summary feature of this family, and LLY-WYZ-7_1 would be labeled as "Additional" genome.

"Representative" genomes and "Additional" genomes were listed in the column named "Fig3-Metabolism" in the Supplementary Table S2. The list of additional genes could be found in the column named "Fig3-Reasons" in the Supplementary Table S2.

iii) We have added the number of genomes in brackets next to each of the family names in the figure. We also added the description on Line 973-974: "Half-filled parts in snowflake indicate the absence of proteins in the representative genome but present in other genomes from the same family."

Fig 4

Were the expression data mapped back to the genomes? In 4A, you may need to point out that the green are photosynthesis genes and the grey are non-photosynthesis genes. Are there multiple Myxococcota genomes from the same study? Are they expressed evenly? In 4B, what

is the representative genome of the family *Ca. Houyibacteriaceae*?

Response: Thanks for your suggestion. i) We mapped the trimmed metatranscriptomic reads to the coding sequences (CDS) of the six representative *Myxococcota* genomes (LLY-WYZ-15_3, LLY-WYZ-17_1, LLY-WYZ-5_1, LLY-WYZ-10_1, LLY-WYZ-9_1, and LLY-WYZ-13_1.). ii) In the Fig. 5a, we used samples with highly expressed photosynthetic genes (considering both of the prevalence and abundance) to visualize the expression pattern (Response letter Fig. 7). We refined the description of Fig. 5a in the caption on Line 1001-1004: “The expression level (FPKB, fragments per kilobase per billion metatranscriptomic fragments) of photosynthesis genes (green) compared with non-photosynthesis genes (grey) of six representative *Myxococcota* genomes in typical samples with photosynthesis gene prevalence >50% and mean FPKB >3.” iii) *Myxococcota* genomes were recovered from different samples. In metatranscriptomic samples, the expression of the photosynthetic gene may be detected simultaneously in multiple *myxococcota* genomes with varied expression levels (Response letter Fig. 8), e.g., the expression of *pufl* gene from three genomes of the families *Ca. Houyibacteriaceae*, *Nannocystaceae*, and *Polyangiaceae* in the sample “080_DCM_0.8to5” (Response letter Table 5). The genome LLY-WYZ-15_3 of *Ca. Houyibacteriaceae* has the highest prevalence and abundance of PGC genes compared with other genomes. This is reasonable because genomes of other families were all recovered from non-ocean environments, e.g., terrestrial or freshwater habitats. iv) Therefore, we select the genome LLY-WYZ-15_3 as representative of *Ca. Houyibacteriaceae* to visualize the expression data in the Fig. 5b. v) The metatranscriptomic datasets were updated by replacing several published samples without clear reference to samples with clear reference according to scientific literature²⁰⁻²⁴.

Response letter Fig. 7 Typical samples (top 30 samples) for visualization were selected based on the photosynthesis gene prevalence and mean abundance (prevalence >50% and mean FPKB >3).

Response letter Fig. 8 The expression of PGCs of representative genomes from the six families in the samples with highly expressed photosynthetic genes (top 30 samples with photosynthesis gene prevalence >50% and mean FPKB >3).

Response letter Table 5 The expression level of photosynthesis gene (*pufl*) from different families of *Myxococcota* across samples.

Sample	Gene	Genome	PGC-ORF	FPKB (Fragments per Kilobase per Billion metatranscriptomic fragments)
100_DCM_0.22to3	pufL	Houyibacteriaceae-LLY-WYZ-15_3	Houyibacteriaceae-LLY-WYZ-15_3-k141_102864_99	294.95
110_DCM_0.8to2000	pufL	Houyibacteriaceae-LLY-WYZ-15_3	Houyibacteriaceae-LLY-WYZ-15_3-k141_102864_99	142.50
111_DCM_5to20	pufL	Houyibacteriaceae-LLY-WYZ-15_3	Houyibacteriaceae-LLY-WYZ-15_3-k141_102864_99	97.57
109_DCM_0.22to3	pufL	Sandaracinaceae-LLY-WYZ-13_1	Sandaracinaceae-LLY-WYZ-13_1-k141_20852_12	73.54
100_DCM_0.8to2000	pufL	Houyibacteriaceae-LLY-WYZ-15_3	Houyibacteriaceae-LLY-WYZ-15_3-k141_102864_99	67.02
110_SRF_0.8to5	pufL	Houyibacteriaceae-LLY-WYZ-15_3	Houyibacteriaceae-LLY-WYZ-15_3-k141_102864_99	65.58
138_SRF_0.22to3	pufL	Houyibacteriaceae-LLY-WYZ-15_3	Houyibacteriaceae-LLY-WYZ-15_3-k141_102864_99	54.10
137_MES_0.22to3	pufL	Houyibacteriaceae-LLY-WYZ-15_3	Houyibacteriaceae-LLY-WYZ-15_3-k141_102864_99	50.26
041_SRF_0.8to5	pufL	Sandaracinaceae-LLY-WYZ-13_1	Sandaracinaceae-LLY-WYZ-13_1-k141_20852_12	49.45
093_SRF_0.8to2000	pufL	Houyibacteriaceae-LLY-WYZ-15_3	Houyibacteriaceae-LLY-WYZ-15_3-k141_102864_99	49.06
122_MES_0.22to3	pufL	Sandaracinaceae-LLY-WYZ-13_1	Sandaracinaceae-LLY-WYZ-13_1-k141_20852_12	46.79
080_SRF_0.8to5	pufL	Sandaracinaceae-LLY-WYZ-13_1	Sandaracinaceae-LLY-WYZ-13_1-k141_20852_12	45.56
080_DCM_0.8to5	pufL	Houyibacteriaceae-LLY-WYZ-15_3	Houyibacteriaceae-LLY-WYZ-15_3-k141_102864_99	42.24
041_SRF_0.22to1.6	pufL	Sandaracinaceae-LLY-WYZ-13_1	Sandaracinaceae-LLY-WYZ-13_1-k141_20852_12	42.18
111_DCM_5to20	pufL	Sandaracinaceae-LLY-WYZ-13_1	Sandaracinaceae-LLY-WYZ-13_1-k141_20852_12	39.14
051_DCM_0.8to5	pufL	Sandaracinaceae-LLY-WYZ-13_1	Sandaracinaceae-LLY-WYZ-13_1-k141_20852_12	38.17
125_SRF_0.22to3	pufL	Houyibacteriaceae-LLY-WYZ-15_3	Houyibacteriaceae-LLY-WYZ-15_3-k141_102864_99	37.53
124_SRF_0.22to3	pufL	Houyibacteriaceae-LLY-WYZ-15_3	Houyibacteriaceae-LLY-WYZ-15_3-k141_102864_99	32.52
111_SRF_0.22to3	pufL	Houyibacteriaceae-LLY-WYZ-15_3	Houyibacteriaceae-LLY-WYZ-15_3-k141_102864_99	31.73
137_MES_0.22to3	pufL	Sandaracinaceae-LLY-WYZ-13_1	Sandaracinaceae-LLY-WYZ-13_1-k141_20852_12	30.26
041_SRF_0.8to5	pufL	Houyibacteriaceae-LLY-WYZ-15_3	Houyibacteriaceae-LLY-WYZ-15_3-k141_102864_99	29.54
051_DCM_20to180	pufL	Sandaracinaceae-LLY-WYZ-13_1	Sandaracinaceae-LLY-WYZ-13_1-k141_20852_12	28.33
080_DCM_0.8to5	pufL	Sandaracinaceae-LLY-WYZ-13_1	Sandaracinaceae-LLY-WYZ-13_1-k141_20852_12	26.31
041_SRF_0.22to1.6	pufL	Nannocystaceae-LLY-WYZ-10_1	Nannocystaceae-LLY-WYZ-10_1-JADJEE010000012_1_443	24.76
052_SRF_0.8to5	pufL	Sandaracinaceae-LLY-WYZ-13_1	Sandaracinaceae-LLY-WYZ-13_1-k141_20852_12	17.78
052_SRF_0.8to5	pufL	Houyibacteriaceae-LLY-WYZ-15_3	Houyibacteriaceae-LLY-WYZ-15_3-k141_102864_99	17.71
135_SRF_0.8to2000	pufL	Houyibacteriaceae-LLY-WYZ-15_3	Houyibacteriaceae-LLY-WYZ-15_3-k141_102864_99	17.61
100_SRF_0.22to3	pufL	Houyibacteriaceae-LLY-WYZ-15_3	Houyibacteriaceae-LLY-WYZ-15_3-k141_102864_99	12.90
041_SRF_0.8to5	pufL	Nannocystaceae-LLY-WYZ-10_1	Nannocystaceae-LLY-WYZ-10_1-JADJEE010000012_1_443	12.81
138_SRF_5to20	pufL	Houyibacteriaceae-LLY-WYZ-15_3	Houyibacteriaceae-LLY-WYZ-15_3-k141_102864_99	12.77
111_SRF_0.8to5	pufL	Houyibacteriaceae-LLY-WYZ-15_3	Houyibacteriaceae-LLY-WYZ-15_3-k141_102864_99	11.42
080_SRF_0.8to5	pufL	Houyibacteriaceae-LLY-WYZ-15_3	Houyibacteriaceae-LLY-WYZ-15_3-k141_102864_99	10.86
080_DCM_0.8to5	pufL	Polyangiaceae-LLY-WYZ-9_1	Polyangiaceae-LLY-WYZ-9_1-JAEUKE010000153_1_187	10.56
093_SRF_0.8to5	pufL	Sandaracinaceae-LLY-WYZ-13_1	Sandaracinaceae-LLY-WYZ-13_1-k141_20852_12	9.45
093_SRF_0.8to5	pufL	Houyibacteriaceae-LLY-WYZ-15_3	Houyibacteriaceae-LLY-WYZ-15_3-k141_102864_99	9.41
138_SRF_0.22to3	pufL	Sandaracinaceae-LLY-WYZ-13_1	Sandaracinaceae-LLY-WYZ-13_1-k141_20852_12	9.06
123_SRF_0.22to3	pufL	Sandaracinaceae-LLY-WYZ-13_1	Sandaracinaceae-LLY-WYZ-13_1-k141_20852_12	9.02

Line 860: Top highly expressed samples? Are the same levels of expression found in the other genomes?

Response: Here we used samples with highly expressed photosynthetic genes (considering both of the prevalence and abundance) to visualize the expression pattern (Response letter Fig. 7). We refined the description of Fig. 5B in the caption on Line 1003-1007: "...in typical samples with photosynthesis gene prevalence >50% and mean FPKB >3. b, The expression of PGC of representative genome (LLY-WYZ-15_3) from the family *Ca. Houyibacteriaceae* in typical samples." The expressions in other genomes are varied across samples (Response letter Table 6). For example, for the sample "100_DCM_0.22to3", the FPKB for *bchL* of three genomes are about 267.58, 54.50, and 19.56, respectively.

Response letter Table 6 The expression level of photosynthesis gene (*bchL*) from different families of *Myxococcota* across samples.

Sample	Gene	Genome	PGC-DRF	FPKB (Fragments per Kilobase per Billion metatranscriptomic fragments)
100_DCM_0.22to3	bchL	Houyibacteriaceae-LLY-WYZ-15_3	Houyibacteriaceae-LLY-WYZ-15_3-k141_102864_126	267.58
110_DCM_0.8to2000	bchL	Houyibacteriaceae-LLY-WYZ-15_3	Houyibacteriaceae-LLY-WYZ-15_3-k141_102864_126	235.19
111_DCM_5to20	bchL	Houyibacteriaceae-LLY-WYZ-15_3	Houyibacteriaceae-LLY-WYZ-15_3-k141_102864_126	119.45
051_DCM_0.8to5	bchL	Houyibacteriaceae-LLY-WYZ-15_3	Houyibacteriaceae-LLY-WYZ-15_3-k141_102864_126	94.22
122_MES_0.22to3	bchL	Nannocystaceae-LLY-WYZ-10_1	Nannocystaceae-LLY-WYZ-10_1-JADJEE010000012_1_453	84.85
093_SRF_0.8to2000	bchL	Houyibacteriaceae-LLY-WYZ-15_3	Houyibacteriaceae-LLY-WYZ-15_3-k141_102864_126	81.44
137_SRF_0.22to3	bchL	Houyibacteriaceae-LLY-WYZ-15_3	Houyibacteriaceae-LLY-WYZ-15_3-k141_102864_126	80.09
125_SRF_0.22to3	bchL	Houyibacteriaceae-LLY-WYZ-15_3	Houyibacteriaceae-LLY-WYZ-15_3-k141_102864_126	76.76
122_MES_0.22to3	bchL	Houyibacteriaceae-LLY-WYZ-15_3	Houyibacteriaceae-LLY-WYZ-15_3-k141_102864_126	69.13
093_SRF_0.8to5	bchL	Houyibacteriaceae-LLY-WYZ-15_3	Houyibacteriaceae-LLY-WYZ-15_3-k141_102864_126	68.52
122_MES_0.22to3	bchL	Sandaracinaceae-LLY-WYZ-13_1	Sandaracinaceae-LLY-WYZ-13_1-k141_20852_22	58.69
138_SRF_5to20	bchL	Houyibacteriaceae-LLY-WYZ-15_3	Houyibacteriaceae-LLY-WYZ-15_3-k141_102864_126	56.84
100_SRF_0.22to3	bchL	Sandaracinaceae-LLY-WYZ-13_1	Sandaracinaceae-LLY-WYZ-13_1-k141_20852_22	56.52
100_DCM_0.22to3	bchL	Sandaracinaceae-LLY-WYZ-13_1	Sandaracinaceae-LLY-WYZ-13_1-k141_20852_22	54.50
109_DCM_0.22to3	bchL	Houyibacteriaceae-LLY-WYZ-15_3	Houyibacteriaceae-LLY-WYZ-15_3-k141_102864_126	39.18
051_DCM_20to180	bchL	Houyibacteriaceae-LLY-WYZ-15_3	Houyibacteriaceae-LLY-WYZ-15_3-k141_102864_126	38.55
041_SRF_0.22to1.6	bchL	Nannocystaceae-LLY-WYZ-10_1	Nannocystaceae-LLY-WYZ-10_1-JADJEE010000012_1_453	37.84
041_SRF_0.8to5	bchL	Sandaracinaceae-LLY-WYZ-13_1	Sandaracinaceae-LLY-WYZ-13_1-k141_20852_22	35.66
041_SRF_0.8to5	bchL	Nannocystaceae-LLY-WYZ-10_1	Nannocystaceae-LLY-WYZ-10_1-JADJEE010000012_1_453	34.77
138_SRF_0.22to3	bchL	Houyibacteriaceae-LLY-WYZ-15_3	Houyibacteriaceae-LLY-WYZ-15_3-k141_102864_126	32.77
109_DCM_0.22to3	bchL	Nannocystaceae-LLY-WYZ-10_1	Nannocystaceae-LLY-WYZ-10_1-JADJEE010000012_1_453	32.68
100_SRF_0.8to2000	bchL	Houyibacteriaceae-LLY-WYZ-15_3	Houyibacteriaceae-LLY-WYZ-15_3-k141_102864_126	30.76
100_DCM_0.8to2000	bchL	Nannocystaceae-LLY-WYZ-10_1	Nannocystaceae-LLY-WYZ-10_1-JADJEE010000012_1_453	30.54
135_DCM_0.8to2000	bchL	Houyibacteriaceae-LLY-WYZ-15_3	Houyibacteriaceae-LLY-WYZ-15_3-k141_102864_126	29.87
122_MES_0.22to3	bchL	Myxococcaceae-LLY-WYZ-5_1	Myxococcaceae-LLY-WYZ-5_1-JALHMW010000019_1_42	27.52
137_MES_0.22to3	bchL	Nannocystaceae-LLY-WYZ-10_1	Nannocystaceae-LLY-WYZ-10_1-JADJEE010000012_1_453	27.19
124_SRF_0.22to3	bchL	Houyibacteriaceae-LLY-WYZ-15_3	Houyibacteriaceae-LLY-WYZ-15_3-k141_102864_126	25.29
041_SRF_0.22to1.6	bchL	Sandaracinaceae-LLY-WYZ-13_1	Sandaracinaceae-LLY-WYZ-13_1-k141_20852_22	22.32
111_SRF_0.22to3	bchL	Houyibacteriaceae-LLY-WYZ-15_3	Houyibacteriaceae-LLY-WYZ-15_3-k141_102864_126	21.67
110_SRF_0.8to5	bchL	Houyibacteriaceae-LLY-WYZ-15_3	Houyibacteriaceae-LLY-WYZ-15_3-k141_102864_126	19.82
080_SRF_0.8to5	bchL	Houyibacteriaceae-LLY-WYZ-15_3	Houyibacteriaceae-LLY-WYZ-15_3-k141_102864_126	19.78
052_SRF_0.8to5	bchL	Nannocystaceae-LLY-WYZ-10_1	Nannocystaceae-LLY-WYZ-10_1-JADJEE010000012_1_453	19.63
122_SRF_0.22to3	bchL	Houyibacteriaceae-LLY-WYZ-15_3	Houyibacteriaceae-LLY-WYZ-15_3-k141_102864_126	19.59
100_DCM_0.22to3	bchL	Nannocystaceae-LLY-WYZ-10_1	Nannocystaceae-LLY-WYZ-10_1-JADJEE010000012_1_453	19.56

Fig 5
Lifestyle
Response: Revised.

Fig S1
In the legend, the completeness is 32.4% to 100% and the contamination is 0% to 8.4%, however in Supplementary Table 1, the contamination is up to 9.33%. It also looks like there are genomes in the tree that have even more contamination. It's difficult to see the classes of interest in this figure. I strongly suggest creating trees of the classes of interest and only including the type species for the other genera plus two or three extra genomes. The genome with the long branch is a little worrying. I would remove that genome. Is it because it's very incomplete? Instead of using unphototroph, I would use non-phototroph. How did you include the 16S rRNA genes? It's very difficult to determine how large the circles are. I would remove the different sizes from the legend.

Response: Thanks for the suggestion. i) We re-visualized Fig. S1 by combining the phylogenetic tree and AAI and ANI analyses. The representative genomes used in this figure were dereplicated and re-selected based on genome quality. In addition, reference genomes and their completeness and contamination which were evaluated by CheckM and CheckM2 were listed in the Supplementary Table S4. ii) We also created trees of the classes of interest in the Fig. S2 (*Myxococcia*) and Fig. S3 (*Polyangia*) following your suggestion to only include the type species for the other genera plus two or three extra genomes. The genome numbers of *Kuafubacteria* are so limited that we did not construct a tree for this class, but the phylogeny can be determined from the Fig. S1. Reference genomes were listed in the Supplementary Table S5. iii) The genome with the long branch was removed from the analysis because of the

low completeness. iv) We removed the typo “unphototroph”. v) We have deleted the wrong description of “rRNA genes”, which we originally meant ribosomal RNA protein sequences. vi) The circles with different sizes from the legend were removed.

Fig S2

The AAI between many of the Houyibacteriaceae is 99-100%. I would consider these MAGs to be the same strain and would de-replicate them at least at the 95% ANI level. The same goes for some of the MAGs in the Nannocystaceae. It would be good to also include a phylogenetic tree next to the names so that we can also visually see how close they are. I now see in Table S1 that the MAGs have been dereplicated but you don't mention this in the main text. Are all the analyses performed on all the MAGs or the dereplicated ones?

Response: Thanks for the valuable comment. i) We dereplicated genomes, and use the dRep genomes for AAI plot (Supplementary Figure S1). ii) We also added a phylogenetic tree next to the names of each MAG. iii) We added relevant description on Line 90-92: “We clustered these 32 MAGs on the basis of 95% whole-genome average nucleotide identities (ANIs) to uncover species-level diversity, revealing 18 candidate species-level taxa”. iv) The dRep result was used for the analyses of PGC (Fig. 2c, Fig. S4, S8), metabolisms (Fig. 3), gene expressions (Fig. 4 and Fig. 5), AAI and ANI (revised Fig. S1). In order to provide more detailed information, the sequences of all MAGs were also used in the phylogenetic trees (Fig. 1, Fig. S2, S3, S5, S6, S9, S10) and genome comparisons (Fig. 2ab and Fig. S7).

Fig S3

It's difficult to know what the green squares were unless you zoom into the document at 300%. When I first looked at it I couldn't see green in the legend. I'm not sure if there's another way you can show the numbers that will be easier to see. Is dark green 1 copy and light green 2 copies?

Response: Thanks for this suggestion. In the revised manuscript, we have deleted this part.

Fig S4

You may want to check that your supplementary files are in order. Fig S4 is first mentioned in line 198 but Fig S5 and Fig S6 are mentioned on line 121.

Response: Checked.

Fig S5

I would remove the legend for the bootstraps as it's difficult to see how large the 0.9, 0.93, 0.95, 0.98 and 1 circles are.

Response: Thanks for the suggestion. we have simplified the legend for the bootstraps.

Fig S6

1,201. Where are the Betaproteobacteria and why are they in brackets?

Response: Checked. *Betaproteobacteria* has been classified as a clade of *Gammaproteobacteria* as shown in the GTDB r207. We have clarified the description in the caption in the revised Fig. S5: “*Gammaproteobacteria* (including *Betaproteobacteria*)”. We also clarified the description on Line 931-932: “*Betaproteobacteria* shared same color with *Gammaproteobacteria* based on the taxonomy in GTDB r207”.

Fig S7

It took me a little while to understand what the different colours of the branches are. It maybe best to remove the brackets.

Response: Removed.

Fig S8

paper80

Response: Corrected.

Table S2

Maybe to be consistent, just have the family name and the species name and remove the f__ for family. You should remove LLY-WYZ-19_1. Based on its completeness, this is probably why it's missing a lot of genes in the metabolism analysis.

Response: Thanks for the suggestion, we removed the "f__" for clarity. We also removed the genome (LLY-WYZ-19_1) for its low completeness.

References:

- 1 Sutton, D., Livingstone, P. G., Furness, E., Swain, M. T. & Whitworth, D. E. Genome-Wide Identification of Myxobacterial Predation Genes and Demonstration of Formaldehyde Secretion as a Potentially Predation-Resistant Trait of *Pseudomonas aeruginosa*. *Frontiers in microbiology* **10**, 2650, doi:10.3389/fmicb.2019.02650 (2019).
- 2 Pasternak, Z. *et al.* By their genes ye shall know them: genomic signatures of predatory bacteria. *The ISME journal* **7**, 756-769, doi:10.1038/ismej.2012.149 (2013).
- 3 Mu, D. S. *et al.* Bradymonabacteria, a novel bacterial predator group with versatile survival strategies in saline environments. *Microbiome* **8**, 126, doi:10.1186/s40168-020-00902-0 (2020).
- 4 Murphy, C. L. *et al.* Genomes of Novel Myxococcota Reveal Severely Curtailed Machineries for Predation and Cellular Differentiation. *Appl Environ Microbiol* **87**, e0170621, doi:10.1128/AEM.01706-21 (2021).
- 5 Waite, D. W. *et al.* Proposal to reclassify the proteobacterial classes Deltaproteobacteria and Oligoflexia, and the phylum Thermodesulfobacteria into four phyla reflecting major functional capabilities. *Int J Syst Evol Microbiol* **70**, 5972-6016, doi:10.1099/ijsem.0.004213 (2020).
- 6 Kanehisa, M., Sato, Y. & Morishima, K. BlastKOALA and GhostKOALA: KEGG Tools for Functional Characterization of Genome and Metagenome Sequences. *J Mol Biol* **428**, 726-731, doi:10.1016/j.jmb.2015.11.006 (2016).
- 7 Treuner-Lange, A. *et al.* PilY1 and minor pilins form a complex priming the type IVa pilus in *Myxococcus xanthus*. *Nat Commun* **11**, 5054, doi:10.1038/s41467-020-18803-z (2020).
- 8 Nan, B. *et al.* Flagella stator homologs function as motors for myxobacterial gliding motility by moving in helical trajectories. *Proc Natl Acad Sci U S A* **110**, E1508-1513, doi:10.1073/pnas.1219982110 (2013).
- 9 Nguyen, L. T., Schmidt, H. A., von Haeseler, A. & Minh, B. Q. IQ-TREE: a fast and effective stochastic algorithm for estimating maximum-likelihood phylogenies. *Mol Biol Evol* **32**, 268-274, doi:10.1093/molbev/msu300 (2015).
- 10 Kishino, H. & Hasegawa, M. Evaluation of the maximum likelihood estimate of the evolutionary tree topologies from DNA sequence data, and the branching order in hominoidea. *J Mol Evol* **29**, 170-179, doi:10.1007/BF02100115 (1989).
- 11 Shimodaira, H. & Hasegawa, M. CONSEL: for assessing the confidence of phylogenetic tree selection. *Bioinformatics* **17**, 1246-1247, doi:10.1093/bioinformatics/17.12.1246 (2001).
- 12 Paoli, L. *et al.* Biosynthetic potential of the global ocean microbiome. *Nature* **607**, 111-118,

- doi:10.1038/s41586-022-04862-3 (2022).
- 13 Mendes-Soares, H. & Velicer, G. J. Decomposing predation: testing for parameters that correlate with predatory performance by a social bacterium. *Microb Ecol* **65**, 415-423, doi:10.1007/s00248-012-0135-6 (2013).
- 14 Pham, V. D., Shebelut, C. W., Diodati, M. E., Bull, C. T. & Singer, M. Mutations affecting predation ability of the soil bacterium *Myxococcus xanthus*. *Microbiology (Reading)* **151**, 1865-1874, doi:10.1099/mic.0.27824-0 (2005).
- 15 Livingstone, P. G., Morphew, R. M. & Whitworth, D. E. Myxobacteria Are Able to Prey Broadly upon Clinically-Relevant Pathogens, Exhibiting a Prey Range Which Cannot Be Explained by Phylogeny. *Frontiers in microbiology* **8**, 1593, doi:10.3389/fmicb.2017.01593 (2017).
- 16 Shilo, M. Lysis of blue-green algae by myxobacter. *J Bacteriol* **104**, 453-461, doi:10.1128/jb.104.1.453-461.1970 (1970).
- 17 Berleman, J. E., Chumley, T., Cheung, P. & Kirby, J. R. Rippling is a predatory behavior in *Myxococcus xanthus*. *J Bacteriol* **188**, 5888-5895, doi:10.1128/JB.00559-06 (2006).
- 18 Bull, C. T., Shetty, K. G. & Subbarao, K. V. Interactions Between Myxobacteria, Plant Pathogenic Fungi, and Biocontrol Agents. *Plant Dis* **86**, 889-896, doi:10.1094/PDIS.2002.86.8.889 (2002).
- 19 Pallen, M. J., Telatin, A. & Oren, A. The Next Million Names for Archaea and Bacteria. *Trends Microbiol* **29**, 289-298, doi:10.1016/j.tim.2020.10.009 (2021).
- 20 Zayed, A. A. *et al.* Cryptic and abundant marine viruses at the evolutionary origins of Earth's RNA virome. *Science* **376**, 156-162, doi:10.1126/science.abm5847 (2022).
- 21 Dominguez-Huerta, G. *et al.* Diversity and ecological footprint of Global Ocean RNA viruses. *Science* **376**, 1202-1208, doi:10.1126/science.abn6358 (2022).
- 22 Salazar, G. *et al.* Gene Expression Changes and Community Turnover Differentially Shape the Global Ocean Metatranscriptome. *Cell* **179**, 1068-1083 e1021, doi:10.1016/j.cell.2019.10.014 (2019).
- 23 Carradec, Q. *et al.* A global ocean atlas of eukaryotic genes. *Nat Commun* **9**, 373, doi:10.1038/s41467-017-02342-1 (2018).
- 24 Tara Oceans Consortium, Coordinators; Tara Oceans Expedition, Participants: Registry of all samples from the Tara Oceans Expedition (2009-2013). PANGAEA, <https://doi.org/10.1594/PANGAEA.875582> (2017)

Reviewer #1 (Remarks to the Author):

Here, I detail my second assessment of the submission titled "Globally distributed *Myxococcota* with photosynthesis gene clusters illuminate the origin and evolution of a potentially chimeric lifestyle" by Li and colleagues.

As stated in my original review, the identification and characterisation of complete photosynthetic gene clusters in MAGs taxonomically classifiable as different groups of *Myxococcota* is extremely exciting, and highly original. This is because the myxobacteria are models for bacterial development and social evolution, and many aspects of their life cycles are known in molecular details. Yet, to date, none of its members have been implicated in photosynthesis. As such the work will be highly interesting to a wider audience especially in the fields of microbiology, botany, and biochemistry.

Initially, prior to revision, the draft did not effectively bolster its strong claims about a link between heterotrophic and prototrophic life-styles housed in the same myxobacterial cells. As such the draft was shaky and highly speculative.

Yet, this has changed dramatically with the latest revision. And I am now very positive about the presented revision.

As a brief recap, in my first assessment I expressed both clear praise but also strong reservation regarding the interpretation and framing of their results.

In brief, I warned that the following problems need considerable attention and revision:

- the problematic interpretation of active photosynthesis from expression data alone
- the shaky conceptualisation of and problematic lines of evidence for linking predation with photosynthesis
- the general ignorance toward well-established myxobacterial biology

However, the authors now more than adequately addressed the problems and short-comings above in their extensive and thorough revisions, by:

- successfully demonstrating that key genes in both photosynthesis and sociality are functional in a heterologous expression model system using elegant, experimental tests,
- providing more careful and adequate interpretation of their results in light of known myxobacterial biology, especially by toning down on the speculative link between predation and photosynthesis (they also scratched the problematic predation gene analysis),
- addressing the important caveats provided by the reviewers, including the correct use of taxonomic nomenclature with adequate tools, and otherwise problematic conclusions.

In conclusion, I can recommend the draft for publication with minor revisions, see below.

Minor comments:

I. 7:

Grammar: "[...] *Myxococcota*, which were known for [...]" - please replace with presence form (which ARE known ...)

II. 63-65:

Shaky phrasing: "Heterologous expression of their pigment biosynthesis genes indicate that they can properly integrated into photosynthesis process." – please rephrase adequately, e. g., "Resulting heterologous expression of their pigment biosynthesis genes strongly suggests that these genes can drive photosynthetic processes in their original myxobacterial background." OR

similar.

L. 140:

Technical term: "vertical evolution" – that's not a proper concept, did you mean *vertical inheritance* (see ll. 169 – 170, where you use that expression correctly)?

Figure 1:

- Thanks for taking into account my initial suggestions to improve the visibility of certain features in the circle tree (i.e., added color information on non-photrophic taxa, Fig. 1a, better highlight of all *Myxococcota*).

- one remaining issue: Gamma- and Alphaproteobacteria cluster in a common group in Figure 1b. But the labeling leads to mild confusion and should ideally be changed: currently a connector links the reddish label "Gammaproteobacteria" with the differently colored circle sector (purple) which widely defines "Alphaproteobacteria". The latter label is also connected to the same sector on the bottom of the figures (Fig. 1b). This is not effective as it might lead to confusion, please adequately change the connector and or sector color.

Reviewer #2 (Remarks to the Author):

I have read the rebuttal and I feel that the authors have addressed my concerns in the revision.

Dear Reviewers,

Thank you again for your careful reading and professional suggestions. All comments here have been taken into full consideration during the preparation of the revised manuscript. Below are our point-by-point responses to the comments. Page and line numbers are consistent with the revised manuscript. Responses of reviewers' questions are in blue color, quotations of new texts in the manuscript are in blown color.

Reviewer #1 (Remarks to the Author):

Here, I detail my second assessment of the submission titled "Globally distributed *Myxococcota* with photosynthesis gene clusters illuminate the origin and evolution of a potentially chimeric lifestyle" by Li and colleagues.

As stated in my original review, the identification and characterisation of complete photosynthetic gene clusters in MAGs taxonomically classifiable as different groups of *Myxococcota* is extremely exciting, and highly original. This is because the myxobacteria are models for bacterial development and social evolution, and many aspects of their life cycles are known in molecular details. Yet, to date, none of its members have been implicated in photosynthesis. As such the work will be highly interesting to a wider audience especially in the fields of microbiology, botany, and biochemistry.

Initially, prior to revision, the draft did not effectively bolster its strong claims about a link between heterotrophic and prototrophic life-styles housed in the same myxobacterial cells. As such the draft was shaky and highly speculative.

Yet, this has changed dramatically with the latest revision. And I am now very positive about the presented revision.

As a brief recap, in my first assessment I expressed both clear praise but also strong reservation regarding the interpretation and framing of their results.

In brief, I warned that the following problems need considerable attention and revision:

- the problematic interpretation of active photosynthesis from expression data alone
- the shaky conceptualisation of and problematic lines of evidence for linking predation with photosynthesis
- the general ignorance toward well-established myxobacterial biology

However, the authors now more than adequately addressed the problems and short-comings above in their extensive and thorough revisions, by:

- successfully demonstrating that key genes in both photosynthesis and sociality are functional in a heterologous expression model system using elegant, experimental tests,

-providing more careful and adequate interpretation of their results in light of known myxobacterial biology, especially by toning down on the speculative link between predation and photosynthesis (they also scratched the problematic predation gene analysis),

- addressing the important caveats provided by the reviewers, including the correct use of taxonomic nomenclature with adequate tools, and otherwise problematic conclusions.

In conclusion, I can recommend the draft for publication with minor revisions, see below.

Response: We would like to thank the reviewer again for the positive remarks and the recommendation for the manuscript to be published.

Minor comments:

I. 7:

Grammar: “[...] *Myxococcota*, which were known for [...]” - please replace with presence form (which ARE known ...)

Response: Revised.

II. 63-65:

Shaky phrasing: “Heterologous expression of their pigment biosynthesis genes indicate that they can properly integrated into photosynthesis process.” – please rephrase adequately, e. g., “Resulting heterologous expression of their pigment biosynthesis genes strongly suggests that these genes can drive photosynthetic processes in their original myxobacterial background.” OR similar.

Response: Thanks for the suggestion. We modified the description on Line 62-64: “Resulting heterologous expression of their pigment biosynthesis genes strongly suggests that these genes can drive photosynthetic processes in their original myxobacterial background.”

L. 140:

Technical term: “vertical evolution” – that’s not a proper concept, did you mean *vertical inheritance* (see II. 169 – 170, where you use that expression correctly)?

Response: Thanks for the suggestion. We revised the “vertical evolution” and “vertical evolutionary history” to “vertical inheritance” on Line 8, Line 65, and Line 140.

Figure 1:

- Thanks for taking into account my initial suggestions to improve the visibility of certain features in the circle tree (i.e., added color information on non-photosynthetic taxa, Fig. 1a, better highlight of all *Myxococcota*).

- one remaining issue: Gamma- and Alphaproteobacteria cluster in a common group in Figure 1b. But the labeling leads to mild confusion and should ideally be changed: currently a connector links the reddish label “Gammaproteobacteria” with the differently colored circle sector (purple) which widely defines “Alphaproteobacteria”. The latter label is also connected to the same sector on the bottom of the figures (Fig. 1b). This is not effective as it might lead to confusion, please adequately change the connector and or sector color.

Response: We thank the reviewer for the valuable comments. We modified the Figure 1 as the

reviewer suggested by changing the connector to dashed lines to distinguish the *Alphaproteobacteria* and *Gammaproteobacteria*.

Reviewer #2 (Remarks to the Author):

I have read the rebuttal and I feel that the authors have addressed my concerns in the revision.

Response: We would like to thank the reviewer again for the constructive comments on the submitted version and the positive remarks on the revisions we made.